# Identification of proximal SUMO-dependent interactors using SUMO-ID

Orhi Barroso-Gomila [1], Fredrik Trulsson[2], Veronica Muratore[1], Iñigo Canosa[1], Laura Merino-Cacho [1], Ana Rosa Cortazar[1,3], Coralia Pérez [1], Mikel Azkargorta[1,4,5], Ibon Iloro[1,4,5], Arkaitz Carracedo [1,3,6,7], Ana M. Aransay [1,4], Felix Elortza [1,4,5], Ugo Mayor [6,7], Alfred C. O. Vertegaal[2], Rosa Barrio [1✉] & James D. Sutherland [1✉]

The fast dynamics and reversibility of posttranslational modifications by the ubiquitin family pose significant challenges for research. Here we present SUMO-ID, a technology that merges proximity biotinylation by TurboID and protein-fragment complementation to find SUMO-dependent interactors of proteins of interest. We develop an optimized split-TurboID version and show SUMO interaction-dependent labelling of proteins proximal to PML and RANGAP1. SUMO-dependent interactors of PML are involved in transcription, DNA damage, stress response and SUMO modification and are highly enriched in SUMO Interacting Motifs, but may only represent a subset of the total PML proximal proteome. Likewise, SUMO-ID also allow us to identify interactors of SUMOylated SALL1, a less characterized SUMO substrate. Furthermore, using TP53 as a substrate, we identify SUMO1, SUMO2 and Ubiquitin preferential interactors. Thus, SUMO-ID is a powerful tool that allows to study the consequences of SUMO-dependent interactions, and may further unravel the complexity of the ubiquitin code.

[1] Center for Cooperative Research in Biosciences (CIC bioGUNE), Basque Research and Technology Alliance (BRTA), Bizkaia Technology Park, Building 801 A, 48160 Derio, Spain. [2] Cell and Chemical Biology, Leiden University Medical Center (LUMC), 2333 ZA Leiden, The Netherlands. [3] CIBERONC, Instituto de Salud Carlos III, C/ Monforte de Lemos 3-5, Pabellón 11, Planta 0, 28029 Madrid, Spain. [4] CIBERehd, Instituto de Salud Carlos III, C/ Monforte de Lemos 3-5, Pabellón 11, Planta 0, 28029 Madrid, Spain. [5] ProteoRed-ISCIII, Instituto de Salud Carlos III, C/ Monforte de Lemos 3-5, Pabellón 11, Planta 0, 28029 Madrid, Spain. [6] Ikerbasque, Basque Foundation for Science, 48011 Bilbao, Spain. [7] Biochemistry and Molecular Biology Department, University of the Basque Country (UPV/EHU), E-48940 Leioa, Spain. ✉email: rbarrio@cicbiogune.es; jsutherland@cicbiogune.es

Ubiquitin-like (UbL) proteins belong to a superfamily of small proteins that attach covalently to target substrates in a transient and reversible manner. The UbL family includes Small Ubiquitin-like Modifiers (SUMOs). The mammalian SUMO family consists of at least three major *SUMO* paralogues (*SUMO1,-2,-3*). Human SUMO2 and SUMO3 share 97% sequence identity, whereas they share 47% of sequence identity with SUMO1[1]. Protein SUMOylation is a rigorously regulated cycle involving an enzymatic machinery that acts in a stepwise manner. Briefly, the C-terminal di-glycine motif of mature SUMOs mediates modification of target lysines in substrates through the sequential action of E1 SUMO-activating enzyme SAE1/SAE2, E2 conjugating enzyme UBC9 and SUMO E3 ligases[2]. If required, SUMO as well as the substrate can be recycled by the action of sentrin-specific proteases (SENPs) that cleave the isopeptide bond. Like Ub, SUMO has internal lysines that can be further modified, extended as SUMO chains, modified by Ub chains to target degradation, or even modified by smaller moieties, like acetyl groups[3–5]. Together, these constitute the concept of the SUMO code and the ongoing challenge is to understand how these modifications drive distinct substrate outcomes and cellular fates.

SUMO plays crucial roles in nuclear processes underlying health and disease such as the DNA damage response, cell cycle regulation, transcription and proteostasis[6]. SUMO is known to control vital biological processes including development[7] and cholesterol homeostasis[8]. Improvements in mass spectrometry technology have led to the identification to date of more than 40,700 SUMO sites within 6,700 SUMO substrates[9]. While cell-wide proteomics approaches can help to understand global SUMO signaling[10], better tools are needed that allow the study of the cause and consequences of particular SUMOylation events for individual substrates.

SUMO can also interact non-covalently with SUMO interacting motifs (SIMs) found in some proteins. SIMs are β strands composed of an hydrophobic core motif that interacts with the hydrophobic residues of the SIM-binding groove of SUMOs to form an intramolecular β-sheet[11]. A well-characterized role of the SUMO-SIM interaction concerns the SUMO-targeted Ub ligases (STUbL). The two described human STUbLs, RNF4 and Arkadia/RNF111, recognize poly-SUMOylated substrates through their SIMs and ubiquitylate them, leading to their proteasomal degradation[12,13]. The SUMO-SIM interaction also plays critical roles in assembling protein complexes: interaction of the SIM1 of Ran Binding Protein 2 (RanBP2) with the SUMOylated version of Ran GTPase-activating protein 1 (RanGAP1) is crucial for the RanBP2/RanGAP1*SUMO1/ UBC9 E3 ligase complex formation[14].

Another intriguing function of SUMO-SIM interaction is the targeting of proteins to Promyelocytic Leukemia Nuclear Bodies (PML NBs). PML NBs are membrane-less ring-like protein structures found in the nucleus. They are bound to the nuclear matrix, make contacts with chromatin fibers[15] and associate with transcriptionally active genomic regions[16]. They consist of a shell composed of PML proteins that surround an inner core in which client proteins localize. Due to the heterogeneity of client proteins, PML NBs have diverse nuclear functions (reviewed in[17,18]). The *PML* gene contains 9 exons and numerous splicing variants. All PML isoforms contain the N terminal TRIpartite Motif (TRIM) that is responsible for PML polymerization and NB formation[19], binding to Arsenic Trioxide (ATO)[20] and may act as an oxidative stress sensor[18]. PML also contains a phospho-SIM located at its exon 7 and shared by most PML isoforms[21]. Almost all PML isoforms contain three putative SUMO sites: K65, K160 and K490. PML SUMOylation is a well-characterized signal for RNF4-mediated ubiquitylation and degradation[22].

Proximity-dependent labeling methods are based on promiscuous labeling enzymes that produce reactive molecules that covalently bind neighbor proteins. Labeled proteins can be then purified and identified using affinity-purification coupled to mass spectrometry methods[23]. Proximity-dependent biotin identification (BioID)[24] uses a promiscuously active *Escherichia coli* biotin ligase (BirA*) generated by a point mutation (R118G) to biotinylate lysines in nearby proteins within an estimated range of 10 nm[25]. By fusing BirA* to specific proteins, BioID efficiently identifies interactors at physiological levels in living cells[26]. It has been extensively used in the Ub field, for instance, to identify substrates of E3 ligases[27,28]. Recently, a more efficient version of BioID, termed TurboID, has been developed[29], being this more suitable for transient protein-protein interaction (PPI) detection. Several studies have developed split-versions and applied protein fragment complementation to BioID and TurboID, where proximal biotinylation is dependent on the proximity of the fusion partners, opening new opportunities for spatial and temporal identification of complex-dependent interactomes[30,31].

To study how SUMOylation and SUMO-SIM interactions can lead to other roles and fates for particular substrates poses particular challenges. SUMOylation occurs transiently and often in a small percentage of a given substrate. Modified proteins can be readily deSUMOylated and SUMO can be recycled and passed to other substrates. SUMO-SIM interactions are also difficult to analyze due to their weak affinity (Kd 1–100 µM). To overcome those technical issues, we developed SUMO-ID, a strategy based on Split-TurboID to identify interactors of specific substrates dependent on SUMO conjugation or interaction. Using PML as a model, we demonstrate that SUMO-ID can enrich for factors that depend on PML-SUMO interaction. Importantly, the identified proteins are represented among proximal interactors of PML identified using full-length TurboID. We also applied SUMO-ID to a less-characterized SUMO substrate, Spalt Like Transcription Factor 1 (SALL1), and identified both known and novel interactors that depend on intact SUMOylation sites in SALL1. Finally, we evaluated the UbL and SUMO-isoform specificity of SUMO-ID and identified SUMO1, SUMO2 and Ubiquitin-preferential interactors of TP53. SUMO-ID is thus a powerful tool to study transient and dynamic SUMO-dependent interaction events. The developed methodology is generic and therefore widely applicable in the Ub and UbL field to identify readers of these modifications for individual target proteins to improve our insight in non-covalent signal transduction by Ub and UbL.

## Results

**Identification of SUMO-dependent interactions: the SUMO-ID strategy.** We posited that Split-TurboID, in which one fragment is fused to SUMO and the complementary fragment to a protein of interest, could identify transient SUMO-dependent interactors (Fig. 1). Upon covalent SUMOylation or non-covalent SUMO-SIM interaction, both fragments are brought together, presumably close enough to allow refolding of the TurboID enzyme. In the presence of biotin, the reconstituted TurboID can then label proximal complexes, which can be purified by streptavidin pull-down and identified by liquid chromatography-mass spectrometry (LC-MS). Due to the high affinity of streptavidin-biotin interaction, harsh cell lysis and stringent washes that significantly reduce unspecific protein binding can be applied. We named this approach SUMO-ID.

**T194/G195 Split-TurboID enables SUMO-ID studies.** We applied the previously described E256/G257 BioID split-point[30] to TurboID, but found it unsuitable for SUMO-ID. While SUMO-dependent reconstitution of the E256/G257 was observed in our pilot experiments, the NTurboID[256]-fusions had a significant background biotinylating activity (Supplementary Fig. 1).

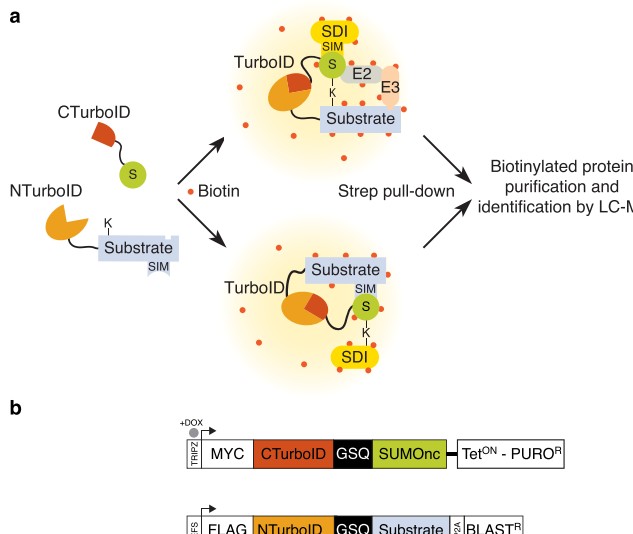

**Fig. 1 Identification of SUMO-dependent interactions: the SUMO-ID strategy.** Schematic representation of the SUMO-ID strategy (**a**) and the constructs used (**b**). BLAST[R] blasticidin resistant cassette, CTurboID C-terminal TurboID, DOX doxycycline, GSQ repetitive linker, NTurboID N-terminal TurboID, P2A 2A self-cleaving peptide, PURO[R] puromycin resistant cassette, S SUMO, SIM SUMO Interacting Motif, SDI SUMO-dependent interactor, SUMOnc SUMO non-cleavable, Strep streptavidin. The sequences of representative constructs are provided in the Source Data file.

This is likely due to residual biotin binding and activation by the intact NTurboID[256] biotin-binding pocket. We examined the BirA structure to identify a TurboID split-point that would yield two completely inactive fragments (see Supplementary Note 1). The biotin-binding pocket of BirA is composed of three β-strands (strands 5, 8 and 9), the N-terminus of helix E and the 110–128 loop (Fig. 2a). We split TurboID at T194/G195, so that the resulting NTurboID[194] fragment (hereafter called N) carries the principal 110–128 biotin-binding loop and the β-strands 5 and 8, while the CTurboID[195] fragment (hereafter called C) carries the β-strand 9 necessary to the formation of the biotin-binding β-sheet.

We tested T194/G195 Split-TurboID for SUMO-ID. C-SUMOs were incorporated into substrates in a very efficient manner (Fig. 2b). N and C alone were catalytically inactive and yielded no biotinylation. Combining N-substrate and C-SUMOs resulted in a high-yield biotinylation activity of TurboID after 16 h of biotin exposure. Modification of N-substrates by C-SUMOs (Fig. 2b, FLAG blot, black arrowheads) and its corresponding biotinylation activity (Fig. 2b, biotin blot, black arrowheads) were efficiently detected, notably in the case of PML protein. Free biotinylated C-SUMOs, that might come from recycling of previously labeled moieties, were observed (Fig. 2b, biotin blot, white squares). We also examined by immunofluorescence, and confirmed that the streptavidin signal recognizing the biotinylated substrates is dependent on fragment complementation (Fig. 2c). Thus, T194/G195 Split-TurboID biotinylation activity is dependent on fragment complementation, with reduced or no leaky biotinylation of the two fragments, so it could be useful for SUMO-ID and for studying other protein-protein interactions.

We applied the rapamycin-inducible dimerization system, based on FKBP (12-kDa FK506-binding protein) and FRB (FKBP-rapamycin-binding domain)[32], which has been used previously to evaluate the PPI dependency of Split-BioID reconstitution[30]. We

fused N and C to FRB and FKBP, respectively, and stably expressed the constructs in HEK293FT cells. We tested short and long rapamycin treatments, together with short biotin-labeling times, to evaluate self-biotinylation activity of the reconstituted TurboID. We observed that biotinylation activity of the reconstituted TurboID correlated well with rapamycin and biotin treatments, showing its dependency on PPI and biotin-labeling times (Fig. 2d). 24 h of rapamycin treatment led to a 15-fold higher FKBP/FRB PPI-dependent biotinylation activity at 2 h of labeling time. Altogether, these data demonstrate that T194/G195 Split-TurboID fragments have low intrinsic affinity and high biotinylation activity at short biotin-labeling times when expressed at low levels, making them suitable for SUMO-ID.

**SUMO-ID detects both covalent and non-covalent SUMO-dependent interactions using short biotin-labeling times.** Interaction of a protein with SUMO can be via covalent SUMOylation or non-covalent SUMO-SIM interaction. We used PML, which can both be SUMOylated and has a well-characterized SIM domain, in conjunction with SUMO wild type (WT) or mutants that lack the C-terminal di-glycine (ΔGG) necessary for covalent conjugation. We used a stable HEK293FT cell line expressing N-PML, into which C-SUMOs were transfected, using short biotin-labeling times (0.5–2 h). We observed that C-SUMO1/2 transfections led to high SUMO-dependent biotinylation activity after only 2 h of biotin treatment (Fig. 3a, biotin blot, black arrowhead). Additionally, ATO treatment, which induces PML SUMOylation, further enhanced the SUMO-dependent biotinylation. With 30 min of biotin treatment, C-SUMO1/2[ΔGG] induced biotinylation of unmodified N-PML, likely through SUMO-SIM interactions (Fig. 3a, biotin blot, white arrowhead). With longer biotin labeling (2 h), biotinylation of endogenous SUMO-modified N-PML was also detected, more strongly in the case of ATO treatment (Fig. 3a, biotin blot, black arrowhead). Biotinylated free C-SUMO1/2[ΔGG] (which are no processable by SENPs and thus are higher in MW than WT C-SUMO1/2) were detected, while the WT counterparts were not biotinylated at 2 h (Fig. 3a, biotin blot, white squares), supporting that recycling of biotinylated SUMOs may be linked to longer labeling times. Indeed, additional experiments showed that appearance of free biotinylated C-SUMOs increased with longer labeling times (Supplementary Fig. 2). Altogether, these results demonstrate that SUMO-dependent biotinylation activity for specific targets, especially at short biotin-labeling times, may be a useful strategy for identifying specific SUMO-dependent interactors of those proteins.

Reduced labeling times and lower expression levels reduced SUMO recycling, but still allowed some degree of recycling (and therefore loss-of-specificity) to occur, so we incorporated two further modifications. First, we designed SUMO isopeptidase-resistant versions of C-SUMOs (SUMO non-cleavable, or SUMOnc, Fig. 1b)[33]. This would avoid the unspecificity derived from recycling of pre-labeled C-SUMOs. In addition, it could also reduce the target identification derived from SUMO-SIM interactions involving free unincorporated C-SUMOs, since most C-SUMOncs would be incorporated into substrates. Furthermore, the use of C-SUMOncs would increase the efficiency of SUMO-ID while decreasing biotin-labeling times, as C-SUMOncs will remain much longer onto the substrate. The same strategy was applied to Ub (C-Ubnc). Secondly, we transferred non-cleavable C-SUMOs into pTRIPZ, an all-in-one doxycycline-inducible (Dox) lentiviral vector (Fig. 1b). Regulated expression would offset any deleterious effects stemming from non-cleavable SUMO isoforms, and provide useful experimental control (i.e., non-induced *versus* induced). Inducible TRIPZ-C-SUMOnc

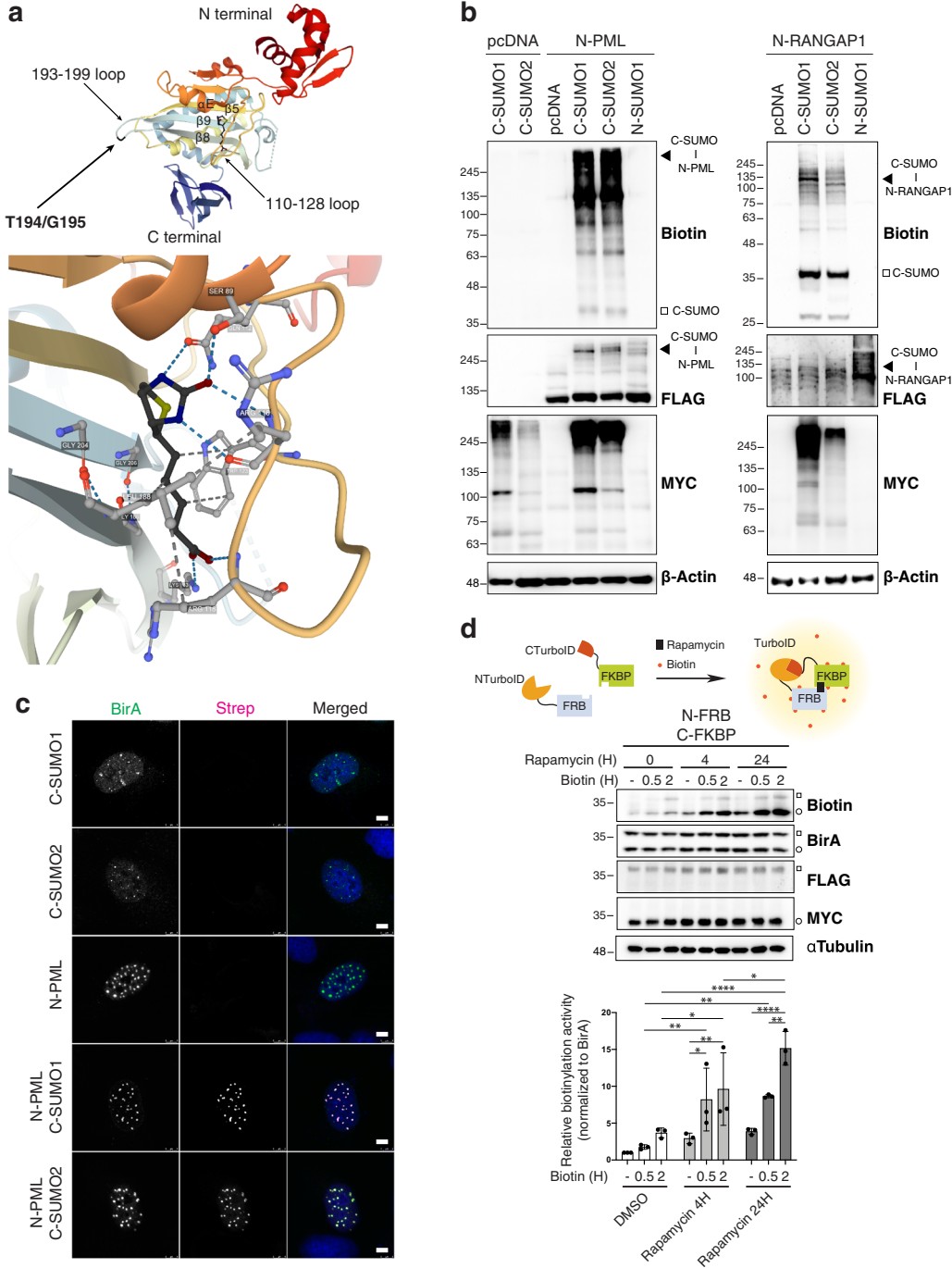

**Fig. 2 T194/G195 Split-TurboID is suitable for SUMO-ID studies. a** Structure of the *E. coli* BirA with bound biotin (PDB ID: 1HXD[80,82]), depicting the T194/G195 split point and the BirA-Biotin interaction. T194/G195 split point breaks the 193-199 loop that connects the biotin-interacting β8 and β9 strands (see Supplementary Note 1). **b** Western blot of HEK293FT cells that were transiently transfected with combinations of the FLAG-N or MYC-C fused to PML, RANGAP1 or SUMO1/2 and treated with 50 μM of biotin for 16 h. Black arrowheads indicate SUMO-ID activity derived from MYC-C-SUMOylated FLAG-N-substrates. White squares indicate biotinylated free MYC-C-SUMOs. Neither FLAG-N nor MYC-C showed any detectable background biotinylating activity. Data are representative of two independent transfection experiments with similar results. Source data are provided in the Source Data file. **c** Immunostainings of transiently transfected U2OS cells treated with 50 μM of biotin for 16 h, showing the fragment-complementation dependency of SUMO-ID and its correct localization within the cell, enriched at PML NBs as expected for SUMOylated PML. Nuclei are stained with DAPI (blue) and biotinylated material with fluorescent streptavidin (Strep, magenta). BirA antibody recognizes both N and C (green). Black and white panels show the single green and magenta channels. Scale bar: 5 μm. Images are representative of two independent transfection experiments performed on cover slips. **d** Western blot of HEK293FT stable cells for N-FRB in combination with C-FKBP, treated or not with 1 μg/mL of rapamycin and 50 μM of biotin at indicated time-points. BirA antibody recognizes both N and C. White squares and circles indicate N-FRB and C-FKBP, respectively. Self-biotinylating activity of the reconstituted TurboID was measured and normalized to expression levels (BirA blot). Bar plots show the mean and standard deviation of three independent experiments. Statistical analyses were performed by 2-way ANOVA: *$p < 0.05$; **$p < 0.01$; ****$p < 0.0001$. Molecular weight markers are shown to the left of the blots in kDa. Source data and the exact $p$ values are provided in the Source Data file.

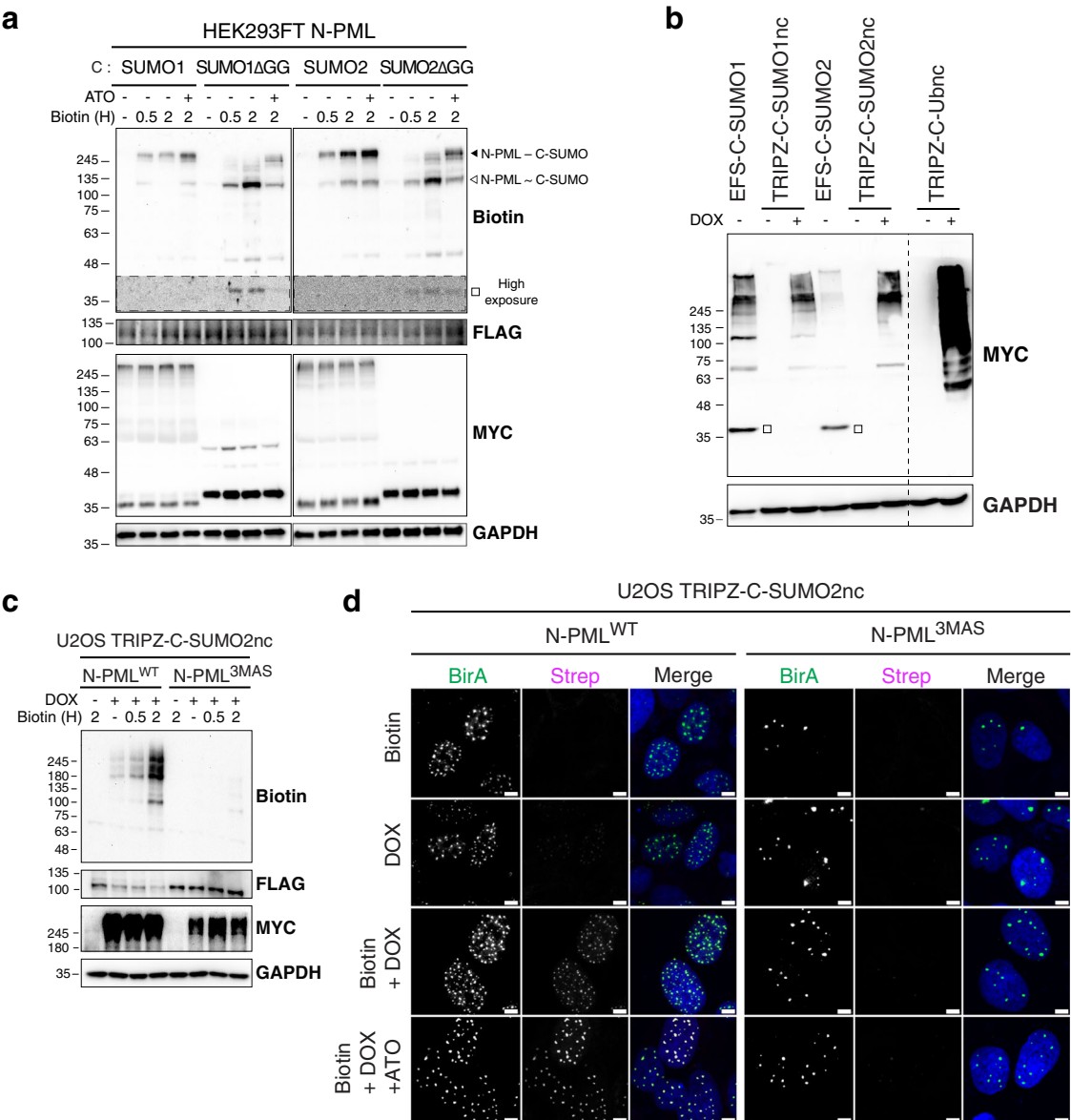

**Fig. 3 SUMO-ID is specific for SUMO-dependent interactions. a** Western blot of HEK293FT FLAG-N-PML stable cell line transfected with different combinations of MYC-C and SUMO$^{WT}$ or SUMO$^{\Delta GG}$. Cells were treated or not with 1 μM of ATO for 2 h and 50 μM of biotin at indicated time points. White square indicates biotinylation of unconjugated MYC-C- SUMO$^{\Delta GG}$. White arrowhead points to SUMO-SIM interaction mediated SUMO-ID. Black arrowhead shows PML-SUMOylation derived SUMO-ID. **b** Western blot of HEK293FT transfected with constitutive MYC-C-SUMO1/2 or doxycycline-inducible and isopeptidase-cleavage resistant (nc) MYC-C-SUMO1/2nc or MYC-C-Ubnc. Doxycycline was added or not at 1 μg/mL for 24 h. White squares point to free/unconjugated MYC-C-SUMOs. Dotted line indicates a cut in the same blot. **c** Western blot of U2OS double stable cell lines for FLAG-N-PML$^{WT}$ or the SUMO/SIM mutant FLAG-N-PML$^{3MAS}$ together with doxycycline-inducible TRIPZ-MYC-C-SUMO2nc. Doxycycline was added or not at 1 μg/mL for 24 h. 50 μM of biotin was added at indicated time-points. PML SUMO-ID showed a high PML/SUMO interaction dependency. **d** Confocal microscopy of the same cells as in **c**, treated or not with doxycycline (1 μg/mL, 24 h), biotin (50 μM, 2 h) and ATO (1 μM, 2 h). Nuclei are stained with DAPI (blue) and biotinylated material with fluorescent streptavidin (Strep, magenta). BirA antibody shows N-PML staining (green). Black and white panels show the single green and magenta channels. Colocalization of the streptavidin and N-PML$^{WT}$ signal is observed within PML NBs, that depends on PML-SUMO interaction. Scale bar: 5 μm. Images are representative of three independent experiments. **a**–**c** are representative of 2 biological replicates with similar results. Molecular weight markers are shown to the left of the blots in kDa. Source data are provided in the Source Data file.

showed enhanced SUMOylation compared to the constitutive WT SUMO versions (Fig. 3b). Free non-incorporated versions of SUMO1/2nc and Ubnc were not detectable (Fig. 3b, MYC blot, white squares). Stable cell populations were established (HEK293FT, U2OS and RPE-1 cells) for each (SUMO1nc, SUMO2nc, Ubnc). Validation of TRIPZ-C-SUMO2nc by WB and immunofluorescence is shown (Supplementary Fig. 3). We then introduced constitutively-expressed N-PML into TRIPZ-C-

SUMO1nc or -SUMO2nc cells and proved that biotinylation occurs in PML NBs as expected, in a doxycycline dependent manner (Supplementary Fig. 4). These data show that the use of regulated SUMOnc versions leads to both high activity and specificity needed for the SUMO-ID approach.

To further validate the specificity of SUMO-dependent biotinylation activity with PML, we generated control cells carrying N-PML$^{3MAS}$, a mutated version of PML lacking the

three principal SUMOylation sites (K65, K160 and K490) and the best-characterized SIM domain. While strong SUMO-ID biotinylation activity was observed with the WT version of PML, this biotinylation activity was completely abrogated in the case of PML[3MAS] (Fig. 3c). This lack of biotinylation activity was specific to SUMO, as ubiquitylation-dependent biotinylation activity was observed in TRIPZ-C-Ubnc / N-PML[3MAS] double stable cell line (Supplementary Fig. 5). N-PML[WT] forms true NBs, while N-PML[3MAS] forms NB-like bodies, as reported previously[34] (Fig. 3d). To confirm that NBs formed by N-PML[WT] are true PML NBs, we generated a YFP-PML cell line by inserting YFP into the endogenous PML locus in U2OS cells (Supplementary Fig. 6), and looked for co-localization of N-PML by confocal microscopy. We observed that N-PML colocalizes well with the endogenous PML at PML NBs (Supplementary Fig. 7a). N-PML[WT] NBs appeared to be smaller and more abundant than NB-like bodies formed by N-PML[3MAS] (Supplementary Fig. 7b). Biotinylation driven by SUMO-ID was observed in NBs containing N-PML[WT], and it was enhanced after 2 h of ATO treatment, but not in the NB-like structures containing N-PML[3MAS] (Fig. 3d). Thus, these results show that SUMO-ID biotinylation activity is dependent on substrate-SUMO interaction.

**SUMO-ID identifies SUMO-dependent interactors of PML**. Since PML NBs are known hubs of SUMO-dependent signaling[17,18], we wondered which interactions in NBs via PML are SUMO-dependent, so we performed SUMO-ID using N-PML[WT] compared to N-PML[3MAS], each combined with TRIPZ-C-SUMO2nc. Biotinylated proteins were purified by streptavidin pull-down and sequenced by LC-MS (Supplementary Data 1). 59 high-confidence SUMO-dependent interactors of PML were enriched in PML[WT] SUMO-ID compared to PML[3MAS] SUMO-ID (Fig. 4a). Among those, SUMO E3 ligases (PIAS1, PIAS2, PIAS4, TRIM28), transcriptional regulators (TRIM22, TRIM24, TRIM33, GTF2I, IRF2BP2, IFI16, ZNF280B, MED23, MEF2D, SNW1, RPAP3), and DNA repair proteins (RMI1, BLM, SLX4, XAB2) were identified. Of note, PIAS1 is known to induce PML SUMOylation[35] and SUMO-SIM interaction of BLM is necessary for its targeting to PML bodies[36], which highlights the specificity of the SUMO-ID strategy. Of particular interest, GTF2I and IRF2BP2, identified here by SUMO-ID, form fusion proteins with RARA and cause Acute Promyelocytic Leukemia (APL, see Discussion)[37,38]. We validated these two proteins, as well as TRIM33 and UBC9, as SUMO-dependent interactors of PML by WB (Fig. 4b).

STRING networking of SUMO-dependent interactors of PML shows a highly interconnected cluster related to protein SUMOylation, DNA damage response and transcriptional regulation (Fig. 4c), while GO enrichment also highlighted protein SUMOylation and transcriptional regulation, as well as DNA repair and stress response pathways (Fig. 4d; Supplementary Data 2). Collectively, this data show that the SUMO-ID strategy can efficiently identify SUMO-dependent interactors of PML, and that SUMO interaction with PML reinforces essential processes.

**PML SUMO-ID hits localize to PML NBs**. We checked whether some of the SUMO-dependent interactors of PML localize to NBs. We looked for co-localization of selected SUMO-dependent PML interactors in our U2OS YFP-PML cell line by confocal microscopy. Within individual cells, we observed frequent and multiple co-localization events for PIAS4, TRIM24, TRIM33 and UBC9 in PML NBs (Fig. 5 and Supplementary Fig. 8), whereas PIAS2, GTF2I and IRF2BP2 colocalizations were less frequent, suggesting heterogeneity in PML NB composition that may depend on different factors (including, but not limited to SUMOylation density, subnuclear localization, cell cycle stage,

other PTMs, or contrastingly, technical limitations with antibodies or fixations).

**SUMO-dependent interactions are a subset of PML proximal proteome**. PML NBs are membraneless structures thought to behave as phase-separated liquids and with high heterogeneity in composition[39]. These characteristics make their purification very challenging, and no proteomic data are nowadays available. Therefore, to compare the obtained PML SUMO-ID specific sub-proteome with the regular PML interactome, we decided to characterize a comprehensive PML and PML[3MAS] proximity interactome using standard full-length TurboID (FLTbID). We generated stable U2OS cell lines for FLTbID-PML[WT], FLTbID-PML[3MAS] and FLTbID alone, and treated them or not with ATO to induce PML SUMOylation. High-confidence PML proximal proteome was composed of 271 proteins that were enriched in FLTbID-PML[WT] samples compared to FLTbID alone (Fig. 6a, Supplementary Data 3). STRING networking showed a main core cluster composed of 73.6% of the identified proteins (Fig. 6b). The most representative subclusters were composed of (1) RNA splicing and mRNA processing proteins, (2) transcription, RNA biosynthesis and DNA damage response proteins and (3) replication and SUMOylation related proteins. This largely aligned with Gene Ontology (GO) enrichment analysis, which revealed that PML proximal interactors participate in replication, transcription, RNA splicing, DNA damage response, cell cycle regulation, SUMOylation and ubiquitylation, and telomere maintenance, consistent with fact that PML in U2OS regulates the ALT mechanism[40] (alternative lengthening of telomeres; Fig. 6c, Supplementary Data 4).

SUMOylation of PML is thought to be a controlling factor for composition and dynamics of NBs. Are all NB interactions linked to PML dependent on SUMO? To answer this question, we subdivided the PML interactome into SUMO-dependent or -independent interactors, by comparing FLTbID-PML[WT] and FLTbID-PML[3MAS] samples. We observed some proteins that likely localize to PML NBs[41], such as NCOR-1, STAT3, JUN, BRCA2 and HDAC9, were also enriched in TurboID-PML[3MAS], suggesting SUMO-independent targeting to PML NBs (Supplementary Data 3). Importantly, many of SUMO-dependent interactors identified by SUMO-ID are part of SUMO-dependent PML NBs proteome using standard TurboID, including PIAS2, PIAS4, TRIM24, TRIM33 and IRF2BP2 (Fig. 7a; Supplementary Data 3), supporting the validity of SUMO-ID to identify SUMO-dependent interactors. Interestingly, scores of some PML interactors decreased after ATO treatment (TRIM24, TRIM33, SENP5), suggesting that those proteins may rapidly undergo dissociation or degradation in response to PML SUMOylation. We confirmed such effect for TRIM24 by WB (Fig. 7b). Altogether, these data confirm that SUMO-ID identified hits are a subset of the SUMO-dependent PML proximal proteome.

**SUMO-dependent interactors of PML are enriched in SIMs**. We expected that many of the SUMO-dependent PML interactors might do so via SUMO-SIM interactions and, therefore, should contain or be enriched in SIMs. To test this, we designed and executed an in silico SIM enrichment analysis. We generated 1000 random lists of 59 proteins (the size of the SUMO-ID identified protein list) and evaluated the presence of SIMs (see Source Data file). The median of single SIM and multiple SIM presence in the random lists were 45.76% and 23.73%, respectively (Fig. 7c). SUMO-ID identified proteins showed a much higher content of SIMs, with single SIM and multiple SIM presence values of 96.61% and 89.83%, respectively. It is noteworthy that around 83% of the identified SIMs in PML SUMO-ID list were preceded or followed within the first 4 amino-acids by acidic residues (D,

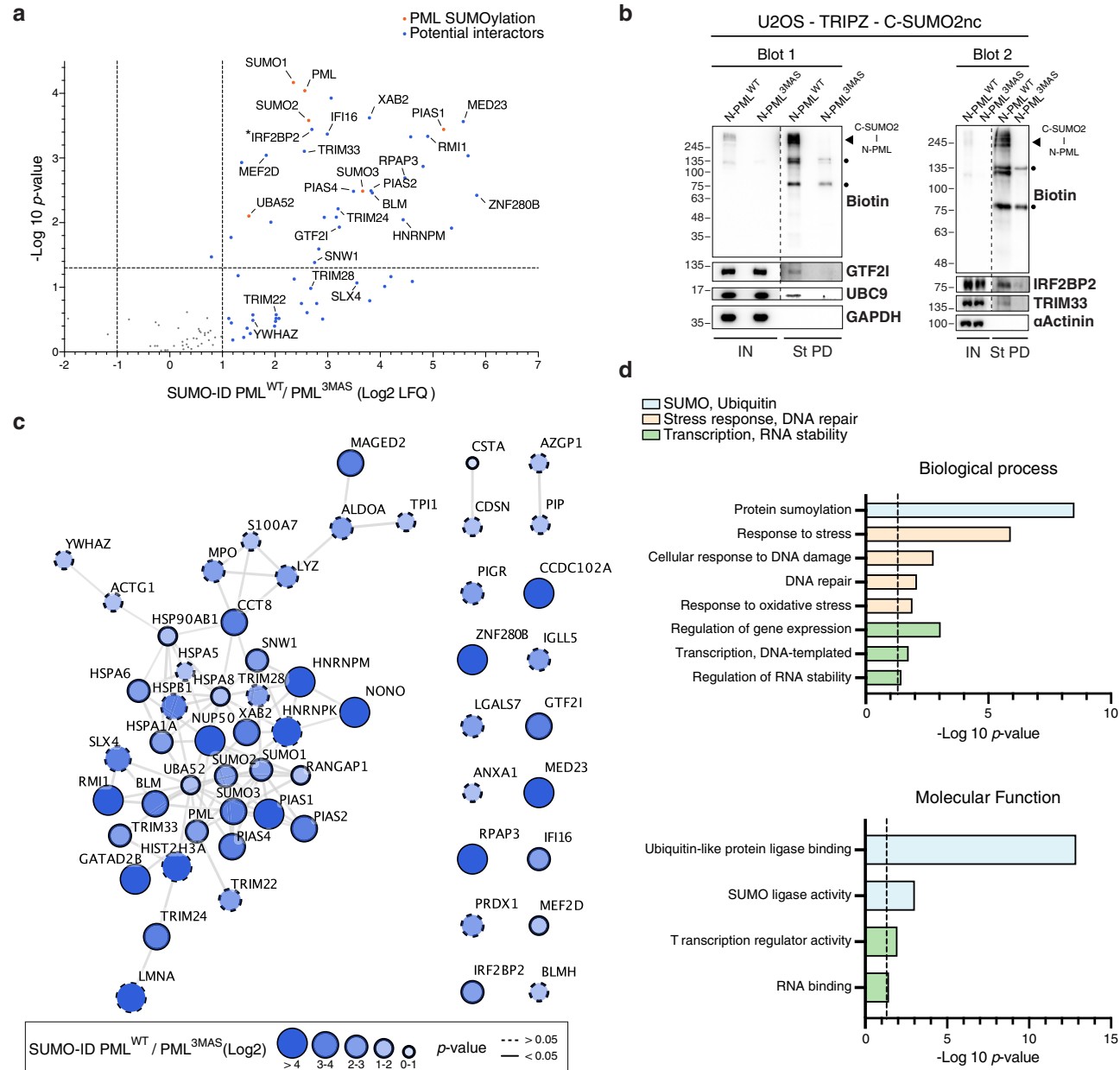

**Fig. 4 SUMO-ID identifies SUMO-dependent interactors of PML. a** Volcano plot of LC-MS analysis comparing streptavidin pull-downs of U2OS double stable cell lines for TRIPZ-MYC-C-SUMO2nc together with FLAG-N-PML^WT or FLAG-N-PML^3MAS. Cells were treated with 1 μg/mL of doxycycline for 24 h and 50 μM of biotin for 2 h. 59 high-confidence SUMO-dependent PML interactors were defined. Asterisk (*) indicates that IRF2BP2 was detected with one peptide but further validated by Western blot and immunofluorescence. Statistical analyses were performed by two-sided Student's *t* test. Data, including proteins identified with 1 peptide, are provided as Supplementary Data 1. **b** Western blot validations of PML SUMO-dependent interactors identified by SUMO-ID in **a**. Blots 1/2 represent two independent experiments. UBC9 was added as an expected positive control. Dots indicate endogenous carboxylases that are biotinylated constitutively by the cell. Black arrowheads point to specific PML SUMO-ID biotinylating activity. Dotted lines indicate different exposures of the same blot. IN: input; St PD: streptavidin pull-down. Molecular weight markers are shown to the left of the blots in kDa. Source data are provided in the Source Data file. **c** STRING network analysis of the 59 SUMO-dependent interactors of PML identified in **a**. A highly interconnected cluster related to protein SUMOylation/ubiquitylation, transcriptional regulation, DNA repair and RNA stability proteins is depicted. Color, transparency and size of the nodes were discretely mapped to the Log2 enrichment value as described. The border line type was discretely mapped to the *p* value as described. **d** Gene ontology analysis of the 59 SUMO-dependent interactors of PML identified in **a**. Biological processes and molecular functions related to SUMOylation/ubiquitylation, stress response, DNA repair, transcription and RNA stability were significantly enriched. Dotted line represents the threshold of the *p* value (0.05). Data are provided as Supplementary Data 2.

E) or a Serine. Since longer proteins are expected to have more SIMs, we then normalized the SIM content with the size of proteins on the lists to obtain the value of "SIMs per thousand of amino acids" (STAA) (see Source Data file). The values obtained with the random lists showed a Gaussian distribution (d'Agostino and Pearson normality test, K2 value 3.836, *p* value 0.15) (Fig. 7d). The median of the values obtained with the random lists was 4.85 (Log2 = 2.28) STAA, while for PML SUMO-ID list was 18.42 (Log2 = 4.20) STAA, which translates to a SIM enrichment value of 3.8 times higher than the random lists (empirical *p*

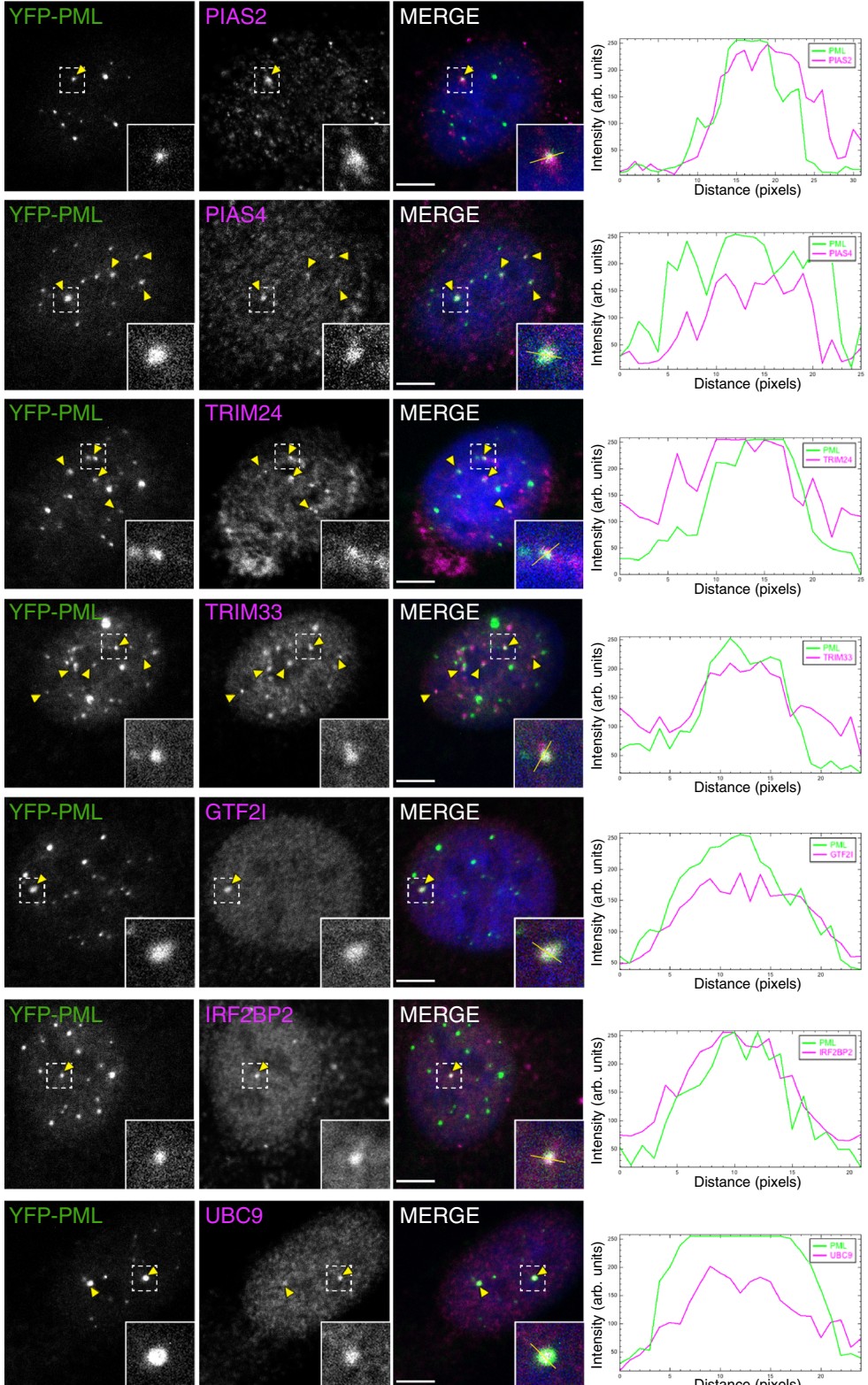

**Fig. 5 SUMO-dependent interactors of PML localize to PML NBs.** Confocal microscopy analysis of PML SUMO-ID identified proteins in U2OS YFP-PML *knock-in* cell line. UBC9 was added as an expected positive control. Yellow arrowheads indicate colocalization events. Dotted line-squares show the selected colocalization events for digital zooming and the signal profile plotting shown to the right. Nuclei are stained with DAPI (blue), YFP-PML is shown in green and the indicated proteins in magenta. Black and white panels show the single green and magenta channels. Images are representative of three independent stainings performed into the same YFP-PML cell line. Further colocalization analysis using the Colocalization Colormap and JACoP is provided as Supplementary Fig. 8. Arb. units Arbitrary units. Scale-bar: 5 μm.

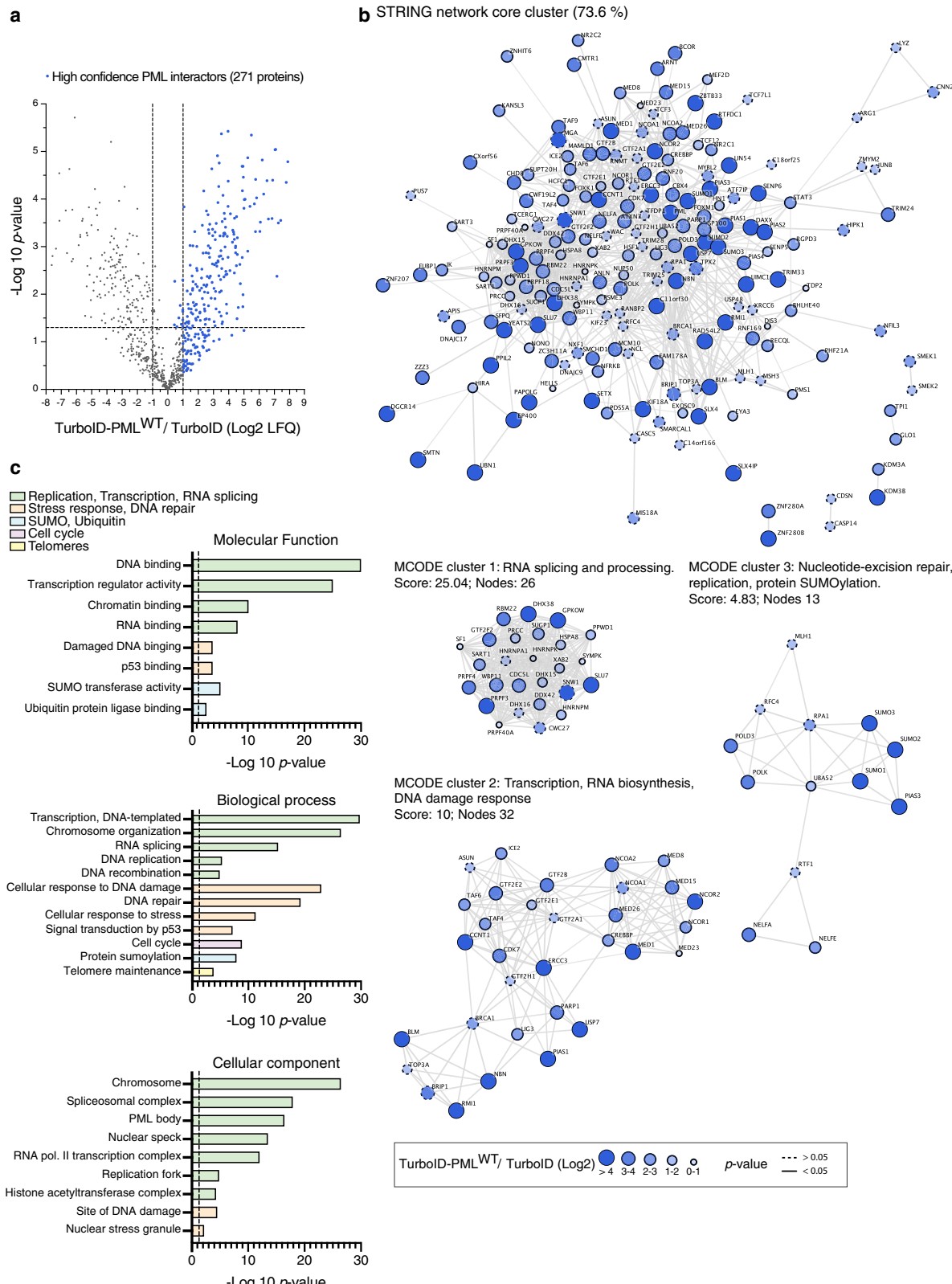

**Fig. 6 Characterization of the whole PML NBs proteome. a** Volcano plot of LC-MS analysis comparing streptavidin pull-downs of U2OS stable cell lines for TurboID-PML^WT or TurboID alone. Cells were treated with 50 µM of biotin for 2 h. High-confidence PML proteome composed of 271 proteins is shown as blue dots. Statistical analyses were performed by two-sided Student's *t* test. Data are provided as Supplementary Data 3. **b** STRING network analysis of the whole PML NBs proteome defined in **a** shows a high interconnected network composed of the 73.6% of the proteins. Highly interconnected sub-clusters were characterized using MCODE. Color, transparency and size of the nodes were discretely mapped to the Log2 enrichment value as described. The border line type was discretely mapped to the *p* value as described. **c** Gene ontology analysis of the whole PML NBs proteome defined in **a**. Depicted biological processes, molecular functions and cellular components were significantly enriched. Dotted line represents the threshold of the *p* value (0.05). Data are provided as Supplementary Data 4.

value < 0.001). These results show that SUMO-dependent interactors of PML are highly enriched in SIMs.

**SUMO-ID identifies interactors of SUMOylated SALL1.** To test the sensitivity and discovery potential of the SUMO-ID, we applied this technique to SALL1, a transcriptional repressor that is SUMOylated[42,43], but of which nothing is known about the causes or consequences of this modification. Using TRIPZ-C–SUMO1nc or SUMO2nc HEK293FT stable cell lines, we introduced N–SALL1$^{WT}$ or SALL1$^{\Delta SUMO}$ (with mutations in 4 major SUMOylation consensus sites) and evaluated SUMO-ID by WB. Efficient SUMO-ID biotinylation activity was observed when using both SUMO1nc and SUMO2nc (Fig. 8a, black arrowhead). N–SALL1$^{WT}$ localizes to the nucleus, forming nuclear bodies with high SUMO-ID activity, and N–SALL1$^{\Delta SUMO}$ also forms aggregates in the cytoplasm (Fig. 8b). Specificity of SALL1 SUMO-ID was confirmed in cells, as biotinylation occurs only in SALL1$^{WT}$ upon doxycycline induction and biotin supplementation. SALL1 SUMO-ID identified potential SUMO-dependent interactors of SALL1 such as the transcription factors TLE3, DACH1/2 and ARID3B, as well as NuRD complex proteins GATAD2A/B, MTA1/2 and RBBP4/7 (Fig. 8c; Supplementary Data 5), already known as SALL1 interactors[44]. We also identified components of the SUMOylation machinery, such as PIAS1. Interestingly, the tyrosine kinase BAZ1B was highly enriched in SALL1$^{\Delta SUMO}$ condition. We confirmed that SUMOylated SALL1 was biotinylated and purified via SUMO-ID (Fig. 8d, black arrowheads) as well as NuRD complex proteins GATAD2B, MTA2 and RBBP4 (Fig. 8d). MCODE subclustering of the STRING interaction network showed a highly interconnected cluster composed of NuRD complex proteins (Fig. 8e) that was also enriched as GO term ($p$ value $2.40{\cdot}10^{-4}$, Supplementary Data 6). Thus, SUMO-ID is sensitive and specific, allowing the study of SUMO-dependent interactors for proteins of interest, opening new avenues of understanding how SUMO can affect their function.

**SUMO-ID identifies UbL and SUMO-paralogue preferential interactors.** In order to further define the specificity of SUMO-ID, we decided to evaluate its capacity to discriminate specific interactors between different UbLs and SUMO paralogues. The cellular tumor antigen TP53 is well known as being modified by SUMO1, SUMO2 and Ub[45,46], making it an attractive candidate to evaluate the specificity of the technique. We thus generated HEK293FT double stable cell lines for N-TP53 together with TRIPZ-C-SUMO1nc, -SUMO2nc or -Ubnc. We validated by WB that N-TP53 was modified with the three different C-UbLnc and confirmed efficient SUMO1-ID, SUMO2-ID and Ub-ID biotinylating activity (Fig. 9a). We thus sequenced by LC-MS the streptavidin pull downs obtained in each of the conditions. We first characterized TP53 SUMO1-ID and SUMO2-ID by comparing them to their non-induced counterparts (Fig. 9b, Supplementary Data 7). We observed that TP53 SUMO1-ID identified many transcriptional regulators and DNA binding proteins (TRIM24, TRIM33, MED23, GTF2I, ZBTB21, ZBTB33, TCERG1, NAB1, ARID3B), SUMO E3 ligases (RANBP2/RAN-GAP1, PIAS1, TRIM28) and DNA repair proteins (PARP1) as SUMO1-dependent interactors of TP53. In addition to those hits, many more proteins were enriched when using SUMO2 (Fig. 9b). Among those, transcriptional regulators (NCOR1, MGA, XAB2, SNW1, GTF3C4, ARID3A, ADNP, PCBP1) and SUMO E3 ligases (PIAS2, PIAS3, ZNF451). Other DNA damage-related proteins were identified as specific SUMO2-dependent interactors of TP53, such as TOP2A, TOP2B, BRCA2, BLM, RPA1, NFRKB and CDKN2AIP (Fig. 9b). Importantly, almost all of the identified hits in SUMO1-ID and SUMO2-ID were enriched while

comparing to TP53 Ub-ID (Fig. 9c), demonstrating the high UbL type-dependent specificity of the technique. TP53 Ub-ID lead to low number but consistent identifications (Fig. 9c): PCNA, BCOR, RNF20, RNF40, RNF220, RFC4, the Ub E1 activating enzyme UBA1 and the deubiquitylating enzyme USP7. We then compared TP53 SUMO1-ID to SUMO2-ID to identify SUMO-paralogue preferences of the interactors (Fig. 9d). Most of the above-described proteins showed preference for SUMO2-dependent interaction, the major hits being SATB1, SATB2 and PCBP1. In the case of SUMO1, two specific interactors were enriched: the SUMO E3 ligase RANBP2 and SUMO1. Thus, SUMO-ID identifies specific interactors for different UbLs and SUMO-paralogues, enabling the understanding of the consequences that each modification might have on the substrate interactions.

**Discussion**
The fast dynamics and reversibility of SUMOylation, and the low affinity of SUMO-SIM interactions pose significant challenges not only for SUMO research, but for respective studies of Ub and other UbLs. The use of His-tagged K0-SUMO or bioSUMO strategy to isolate substrates and map SUMOylation sites has been instrumental to show the widespread presence of this modification in the human proteome[9,43,47]. Direct purification of SUMOylated proteins using immunoprecipitation is a gold standard and can be applied to cells and tissues[48], but is also challenging because SUMOylation might affect a small proportion of low abundance proteins, and perhaps only under certain conditions (e.g., a discrete cell cycle phase or upon DNA damage). Recently, the NanoBiT-based ubiquitin conjugation assay (NUbiCA) was described that uses a split-luciferase approach to allow a quantitative assessment of Ub-modified proteins[49]. Bimolecular fluorescence complementation (BiFC) approaches employ a split fluorescent protein that enables the localization of UbL-modified proteins in yeast or human cells to be monitored[50–52]. If applied to UbLs and substrates, the BiCAP method[53], which allows purification of reconstituted GFP using GFP nanobodies, could likely enrich modified substrates and perhaps interactors. However, none of these approaches captures the dynamic environment of specific UbL-modified proteins, often characterized by weak and transient interactions.

Here we describe SUMO-ID, a powerful technique that allows the study of the causes and consequences of SUMO-dependent interactions for specific proteins of interest. The fast biotinylation activity of TurboID and the specificity obtained with "protein fragment complementation" permit SUMO-ID to specifically biotinylate interactors of substrates in a SUMO-dependent manner. Combined with sensitive proteomic methods, SUMO-ID allows the identification of specific interactors, potentially revealing enzymatic machinery responsible for the SUMOylation as well as interactors that may be stabilized or recruited as a consequence of the modification. Like approaches using BiFC, the subcellular localization of SUMO-modified substrates using SUMO-ID can also be inferred, through the use of fluorescent streptavidin. However, caution should be taken with the mentioned factors in order to maintain specificity, such is the use of non-cleavable forms of UbLs or the application of short biotin-labeling times. This strategy might compromise the identification of SUMO isopeptidases since their binding to SUMOylated substrate is likely affected.

At the core of SUMO-ID is Split-TurboID, which individual halves should ideally have no activity, as with all split-protein approaches. For SUMO-ID, we initially applied the E256/G257 split point described for Split-BioID[30] to the fast-labeling TurboID derivative, but found that the N-terminal half (1–256)

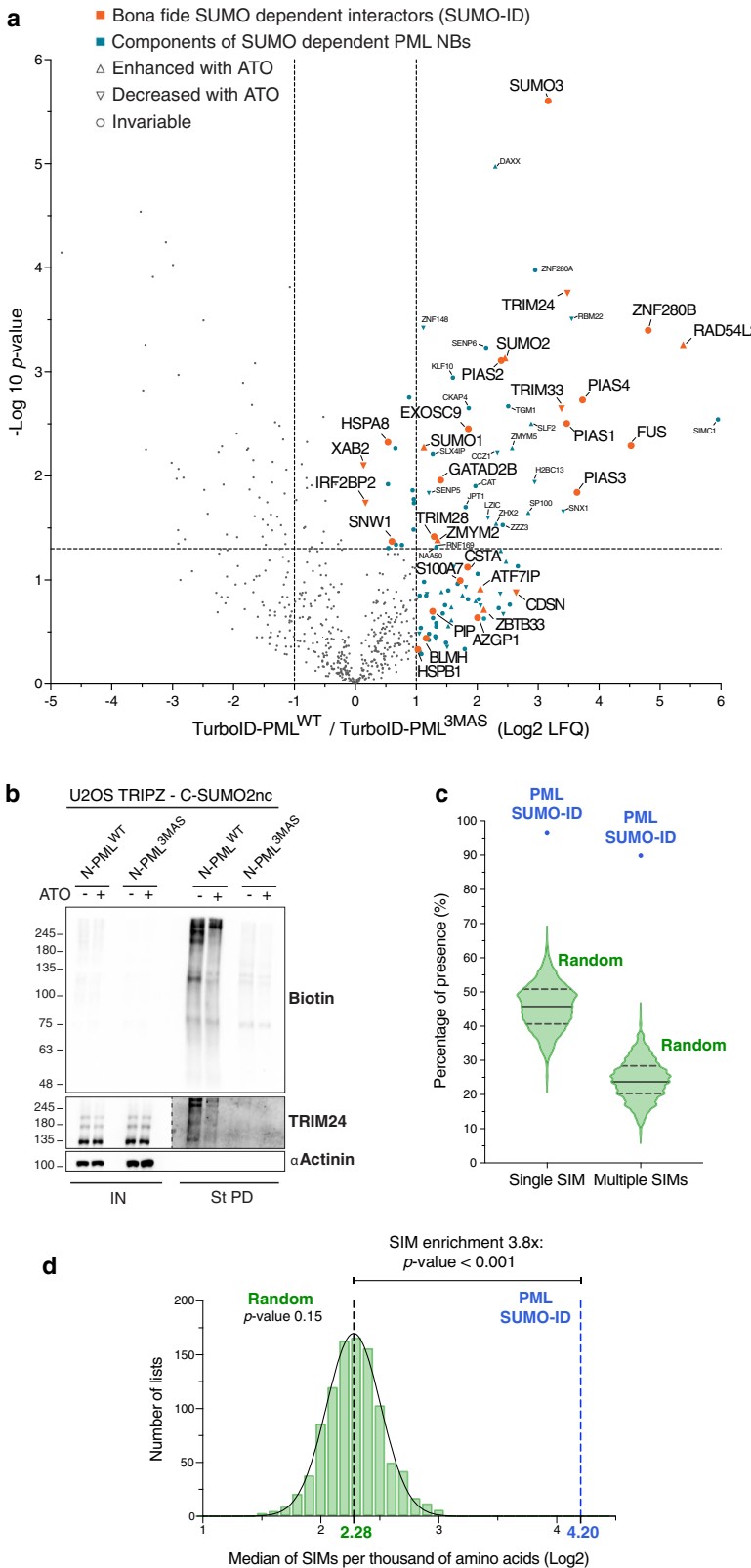

retained substantial biotinylation capacity. We speculate that this is because the biotin pocket is still intact and might allow leaky release of biotinoyl-AMP. Leaky biotinylation of TurboID 1–256 was also observed by Cho et al. in their recently published report on Split-TurboID[31]. Their final design was based on a L73/

G74 split point which showed efficient proximity-dependent reconstitution and biotinylation, but still leaves the biotin pocket intact in C-terminal 74–321 half, opening the possibility of leaky biotinylation during longer labeling times or in stable cell lines. To avoid this, we developed and validated T194/G195 Split-

**Fig. 7 Proteins identified by PML SUMO-ID are a subset of the SUMO-dependent PML NBs proteome and are enriched in SIMs. a** Volcano plot of LC-MS analysis comparing streptavidin pull-downs of U2OS stable cell lines for TurboID-PML^WT or TurboID-PML^3MAS. Cells were treated with 50 μM of biotin for 2 h. Proteins enriched in TurboID alone compared to TurboID-PML^WT were previously removed for the comparison. PML SUMO-ID identified proteins (including 1 peptide identified proteins) are highlighted in orange. LC-MS data on the effect of the ATO treatment (1 μM; 2 h) for TurboID-PML^WT enriched proteins is represented with symbols as described. Statistical analyses were performed by two-sided Student's *t* test. Data are provided as Supplementary Data 3. **b** WB validation of the effect of ATO treatment (1 μM; 2 h) on TRIM24 by PML SUMO-ID. Cells were treated with 1 μg/mL of doxycycline for 24 h and 50 μM of biotin for 2 h. After streptavidin pulldown, decreased levels of SUMO-PML interacting TRIM24 upon ATO treatment is observed. Dotted lines indicate different exposures of the same blot. Data are representative of three independent pull-down experiments. Molecular weight markers are shown to the left of the blots in kDa. IN: input; St PD: streptavidin pull-down. **c** Violin plots comparing the percentage of single SIM and multiple SIM presence in 1000 random lists and PML SUMO-ID list. Horizontal solid and dotted lines represent the median and quartiles (Q1, Q3), respectively. The 1000 random lists contain the same number of proteins (59) as the SUMO-ID list. **d** SIM presence was normalized by the length of the proteins to obtain the value of SIMs per thousand of amino acids (STAA). Gaussian distribution of STAA median values of the random lists was validated (d'Agostino and Pearson normality test, *p* value = 0.15), and PML SUMO-ID SIM enrichment factor with its corresponding empirical *p* value was calculated. The dotted black line represents the median STAA value of random lists. The dotted blue line represents the STAA value of the PML SUMO-ID list. Source data for **b**, **c** and **d** are provided in the Source Data file.

TurboID that separates the β-strands 5 and 8 from the β-strand 9, completely abrogating any residual biotinylation activity of the fragments.

Here we used SUMO-ID to unravel the role of PML SUMOylation in PML NBs function. We identified 59 proteins as SUMO-dependent PML interactors that participate in essential nuclear processes such as protein SUMOylation, transcriptional regulation, DNA repair and stress response. There is growing evidence that PML interaction with SUMO might allow partners to localize into PML NBs through SUMO-SIM interactions[36,54]. We demonstrated that most of the proteins identified by SUMO-ID are indeed part of the proteome of SUMO-dependent PML NBs and that they are enriched in SIMs, suggesting SUMO-SIM interaction dependency. It has been proposed that, after such partner recruitment, proteins might undergo SUMOylation by the PML NB-localized SUMO machinery that reinforces their sequestration[55]. In fact, PML NBs are, together with the nuclear rim, the major targets of active SUMOylation[56]. Our data reinforce this enzyme/substrate co-concentration model as we observed that SUMOylation machinery enzymes (UBC9, PIAS1, PIAS2, PIAS4 and TRIM28) localize to PML NBs in a SUMO-dependent manner and 80% of the SUMO-dependent PML interactors (47 out of 59) are SUMO substrates[9,10].

To compare our list of SUMO-dependent *versus* general interactors of PML, we performed a TurboID assay for PML, with cells alone or treated with ATO, and identified 271 proteins. ATO induces PML NB formation, subsequent PML SUMOylation, partner recruitment and finally PML degradation[22,57]. It is used to treat APL, a type of Acute Myelocytic Leukemia (AML), which is mainly caused by the t(15;17) translocation that fuses PML to RARA. Interestingly, two of our SUMO-ID hits, IRF2BP2 and GTF2I, also form fusion proteins with RARA and cause APL, albeit less commonly than PML fusions[37,38]. We validated that both localize to PML NBs. While many of the SUMO-ID candidates show increased peptide intensity in PML NBs after ATO treatment, we observed that some of them decreased. IRF2BP2 and TRIM24, which has also been linked to AML[58,59], showed reduced levels after ATO treatment, suggesting that they might undergo degradation. In line with this idea, the 11S proteasome components are recruited into mature PML NBs and their localization is enhanced with ATO treatment[60], suggesting that mature PML NBs may also act as proteolytic sites. In fact, inhibition of ubiquitylation accumulates SUMOylated proteins within PML NBs[61], suggesting that many clients that are targeted to PML NBs and that are SUMOylated, might undergo ubiquitylation and degradation. Altogether, these data provide further insight into the role of PML SUMOylation in NB biology and open new ways of looking at the mechanisms of ATO in APL treatment.

The successful application of SUMO-ID to SALL1, a poorly characterized SUMO substrate, illustrates the sensitivity and utility of SUMO-ID. Although SALL1 SUMOylation levels are vanishingly low under physiological conditions, SUMO-ID revealed SUMO-dependent enrichment of the NuRD complex proteins GATAD2A/B, MTA1/2 and RBBP4/7. The association between SALL1, a transcriptional repressor, and the NuRD complex, a repressive histone deacetylase complex, has been previously described[44]. The interaction is mediated by an N-terminal 12 amino acid motif of SALL1[44]. Once recruited, SUMOylation of SALL1 might serve to stabilize the repressor complex via SUMO-SIM interactions, with predicted SIMs present in multiple NuRD complex subunits. As histone SUMOylation is also linked to transcriptional repression[62], SUMO-SIM interactions might further reinforce the SALL1/NuRD complex and drive histone deacetylation at SALL1 targets. In addition, we also found TLE3, DACH1/2 and ARID3B transcription factors as SUMO-dependent interactors of SALL1. TLE3, a transcriptional repressor of the Groucho/TLE family, interacts with HDAC2 (another NuRD complex component) and can regulate acetylation levels[63]. Both TLE3 and the tumor suppressor DACH1 are negative regulators of Wnt signaling[64,65]. Interestingly, SALL1 has been shown to enhance Wnt signaling[66]. Perhaps interaction with SUMOylated SALL1 serves to counteract these negative effects. BAZ1B, a tyrosine-protein kinase that acts as a transcriptional regulator, was highly enriched in SALL1^ΔSUMO compared to SALL1^WT. BAZ1B is highly SUMOylated with at least 5 putative SUMO sites identified (K826, K853, K1043, K1089 and K1107)[9]. Our data suggest that interaction between SUMOylated BAZ1B and SALL1 might be enhanced when SALL1 SUMOylation is inhibited. It would be of potential interest to study if that interaction leads to SALL1 phosphorylation.

We evaluated the capacity of SUMO-ID to discriminate interactors among different UbLs and SUMO paralogues by performing SUMO1-ID, SUMO2-ID and Ub-ID on TP53. We identified many transcriptional regulators, DNA binding proteins, SUMO E3 ligases and DNA damage response proteins as SUMO-dependent interactors of TP53. Interestingly, many of those interactors overlap with those of PML, consistent with the fact that TP53 localizes to PML NBs[17]. While comparing SUMO1-ID to SUMO2-ID, most of the identified proteins showed preferences to SUMO2-dependent interaction except for the SUMO E3 ligase RANBP2 and RANGAP1. This is in line with a recent screening of non-covalent SUMO interactions that showed much more preferential interactors of SUMO2 than SUMO1[67]. Of particular

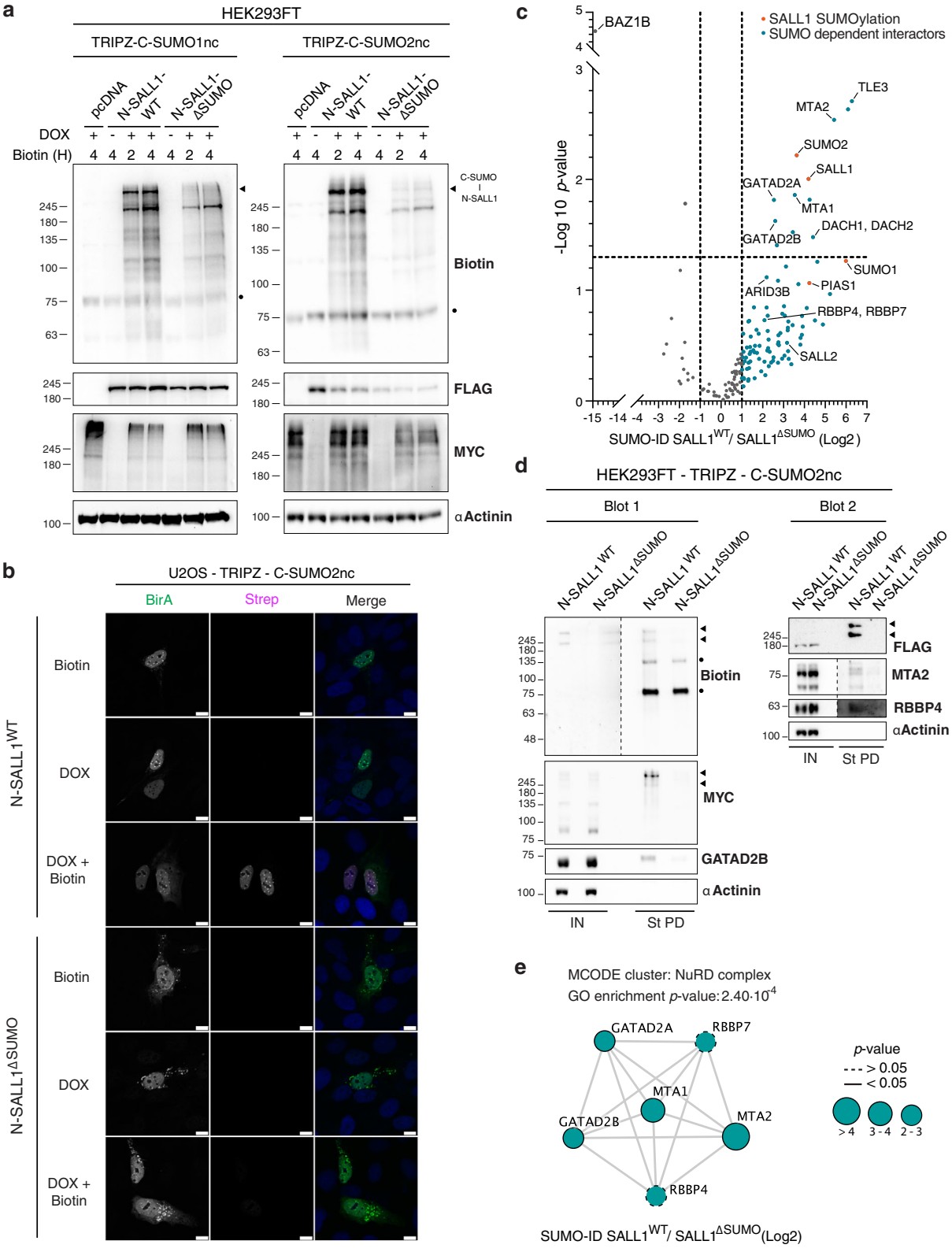

interest, we identified ZNF451, TOP2A and TOP2B as SUMO2-specific interactors of TP53. ZNF451 is a DNA repair factor and poorly characterized SUMO E3 ligase that SUMO2ylates Topoisomerase 2 (TOP2A and TOP2B) and controls cellular responses to TOP2 damage[68]. Importantly, we also identified known interactors of TP53 as SUMO2-dependent specific interactors, such as BRCA2, BLM, RPA1 and DDX5, while the strongest hits

were SATB1/SATB2 and PCBP1. Ub-ID also identified consistent Ub-dependent interactors of TP53 when comparing to SUMO1-ID or SUMO2-ID. Among those, enzymes from the ubiquitin machinery (UBA1 and USP7), as well as potential interesting interactors such as PCNA, and the ubiquitin ligase complex RNF20/RNF40, were enriched. Interestingly, PCNA has been shown to interact with TP53 and regulate its polyubiquitylation

**Fig. 8 SUMO-ID identifies interactors of SUMOylated SALL1. a** WB of HEK293FT stable cell lines for TRIPZ-MYC-C-SUMO1nc/SUMO2nc transfected with FLAG-N-SALL1$^{WT}$ or the SUMO site mutant FLAG-NTurboID$^{194}$-SALL1$^{\Delta SUMO}$. Cells were treated or not with 1 μg/mL of doxycycline for 24 h and 50 μM of biotin at indicated time points. Efficient SALL1 SUMO-ID biotinylating activity was detected for SUMO1nc and SUMO2nc (black arrowhead). Dots indicate endogenous carboxylases that are biotinylated constitutively by the cell. Results are representative of two independent transfection experiments on the same stable cell lines. Source data are provided in the Source Data file. **b** Confocal microscopy of U2OS stable cell line for TRIPZ-MYC-C-SUMO2nc transfected with FLAG-N-SALL1$^{WT}$ or the SUMO site mutant FLAG-N-SALL1$^{\Delta SUMO}$. Cells were treated or not with 1 μg/mL of doxycycline for 24 h and 50 μM of biotin for 4 h. Nuclei are stained with DAPI (blue) and biotinylated material with fluorescent streptavidin (Strep, magenta). BirA antibody shows N-SALL1 staining (green). Black and white panels show the single green and magenta channels. Nuclear colocalization of FLAG-N-SALL1$^{WT}$ and streptavidin signal was observed. Images are representative of three independent transfection experiments performed on cover slips on the same stable cell line. Scale bar: 10 μm. **c** Volcano plot of LC-MS analysis comparing streptavidin pull-downs of HEK293FT TRIPZ-MYC-C-SUMO2nc stable cell line transfected with FLAG-N-SALL1$^{WT}$ or the SUMO site mutant FLAG-N-SALL1$^{\Delta SUMO}$. Cells were treated with 1 μg/mL of doxycycline for 24 h and 50 μM of biotin for 4 h. Potential interactors of SUMOylated SALL1 are depicted. Statistical analyses were performed by two-sided Student's *t* test. Data are provided as Supplementary Data 5. **d** Western blot validations of SUMOylated SALL1 interactors found in **c**. NuRD complex proteins GATAD2B, MTA2 and RBBP4 were confirmed. Black arrowheads point to SUMOylated SALL1 signal. Dots indicate endogenous carboxylases that are biotinylated constitutively by the cell. Dotted lines indicate different exposures of the same blot. Data are representative of three independent transfection experiments. IN input, St PD streptavidin pull-down. Source data are provided in the Source Data file. **e** STRING network analysis of the SALL1 SUMO-ID list and MCODE clustering identifies the NuRD complex as a highly interconnected subcluster. Gene ontology analysis also identified the NuRD complex as an enriched term. Color, transparency and size of the nodes were discretely mapped to the Log2 enrichment value as described. The border line type was discretely mapped to the *p* value as described. Data are provided as Supplementary Data 6. Molecular weight markers are shown to the left of the blots in kDa in **a** and **d**.

and stability together with MDM2[69]. In addition, TP53 mediates the recruitment of RNF20/ RNF40 ubiquitin ligase complex to TP53 target gene loci to regulate transcription through H2B ubiquitylation[70]. Perhaps TP53 ubiquitylation regulates the stability of such complex formation. Thus, SUMO-ID appear to be highly specific, enabling the identification of specific interactors of SUMO1, SUMO2 and ubiquitin-modified substrates.

In summary, we demonstrate here that SUMO-ID, based on the 194/195 Split-TurboID reconstitution, can facilitate the identification of SUMO-dependent interactions with a protein of interest. It has little or no background, with high biotinylation activity when expressed at low levels and with short biotin incubation time. We believe that this technique improves sensitivity and selectivity when applied to infrequent SUMOylation events and low-affinity of SUMO-SIM interactions. This strategy can be applied to other UbL modifications (e.g., Ub-ID shown in Fig. 9), and the 194/195 Split-TurboID may be useful for other applications in cell biological studies. We recommend the use of "SUMO-dead" versions of the substrates as negative controls while performing SUMO-ID studies. Otherwise, the use of ΔGG versions of C-SUMOs could also represent a correct negative control. A non-induced vs induced strategy, or comparing different UbL-IDs would be convenient controls when the SUMO sites of the substrate are not known. We also recommend the use of the GSQ linker between the C and SUMOs as well as between the N and substrates to enable efficient SUMO-ID activity. The length of the linker could be modified depending on the distance of the SUMO site and the N-terminus of the substrate.

## Methods

**Cell culture.** U2OS (ATCC HTB-96) and HEK293FT (Invitrogen) were cultured at 37 °C and 5% CO$_2$ in Dulbecco's modified Eagle Medium (DMEM) supplemented with 10% fetal bovine serum (FBS, Gibco) and 1% penicillin/streptomycin (Gibco). Human telomerase reverse transcriptase immortalized retinal pigment epithelial cells (hTERT-RPE1, ATCC CRL-4000) were cultured in DMEM:F12 (Gibco) supplemented with 10% FBS, 2 mM L-Glutamine and 1% penicillin and streptomycin. Cultured cells were maintained through 20 passages maximum and tested negative for mycoplasma.

**Cloning.** TurboID was a kind gift of A. Ting (Addgene #107171)[29]. PMLIVa$^{WT}$ and PMLIVa$^{3MAS}$ were previously described[21]. SUMO1, SUMO2, Ub, RANGAP1 and UBC9 ORFs were amplified from U2OS cell cDNA by high-fidelity PCR (Platinum SuperFi DNA Polymerase; Invitrogen). A GSQ linker (GGGSSGGGQISYASRG) was placed between the C-terminal part of TurboID and the UbLs as well as between the N-terminal part and the substrates. All constructs were generated by standard cloning or by Gibson Assembly (NEBuilder HiFi

Assembly, NEB) using XL10-Gold bacteria (Agilent). Depending on the construction, plasmid backbones derived from EYFP-N1 (Clontech/Takara), Lenti-Cas9-blast (a kind gift of F. Zhang; Addgene #52962) or TRIPZ (Open Biosystems/Horizon) were used. After assembly, all vectors were validated by sequencing. Additional details for constructs are described in Supplementary Table 1. Oligonucleotides sequences are shown in Supplementary Table 2. The sequences of representative constructs are in the Source Data file. Cloning details about other constructs are available upon request.

**Lentiviral transduction.** Lentiviral expression constructs were packaged in HEK293FT cells using calcium phosphate transfection of psPAX2 and pMD2.G (kind gifts of D. Trono; Addgene #12260, 12259) and pTAT (kind gift of P. Fortes; for TRIPZ-based vectors). Transfection media was removed after 12–18 h and replaced with fresh media. Lentiviral supernatants were collected twice (24 h each), pooled, filtered (0.45 μm), and supplemented with sterile 8.5% PEG6000, 0.3 M NaCl, and incubated 12–18 h at 4 °C. Lentiviral particles were concentrated by centrifugation (1500 × *g*, 45 min, 4 °C). Non-concentrated virus (or dilutions thereof) were used to transduce HEK293FT, and 8× concentrated virus was used for U2OS and hTERT-RPE1 cells. Drug selection was performed as follows: 1 μg/ml puromycin (Santa Cruz) for U2OS and HEK293FT cells, 5 μg/ml for hTERT-RPE1 cells; 5 μg/ml blasticidin (Santa Cruz) for U2OS, HEK293FT and hTERT-RPE1 cells.

**CRISPR-Cas9 genome editing.** Human PML encodes multiple isoforms, but most differ at the 3' end. To target EYFP into the first coding exon, shared by most PML isoforms, an sgRNA target site was chosen (CTGCACCCGCCCGATCTCCG) using Broad institute GPP sgRNA Designer[71]. Custom oligos were cloned into px459v2.0 (a kind gift of F. Zhang; Addgene #62988). A targeting vector was made by amplifying 5' and 3' homology arms using U2OS genomic DNA, as well as the EYFP ORF (see Supplementary Table 2 for details on the oligonucleotides). These fragments were assembled by overlap extension using high-fidelity PCR and the resulting amplicon was TOPO-cloned and sequence-confirmed. Lipofectamine 2000 (Invitrogen) was used to transfect U2OS with linear targeting vector and px459 encoding SpCas9, puromycin resistance, and the PML-targeting sgRNA. 24 h post-transfection, cells were selected for additional 24–36 h with 2 μg/ml puromycin. Cells were plated at low density and clones were examined by fluorescence microscopy. Clones with clear nuclear body signals were manually picked and expanded. YFP-PML insertions and copy number were validated by PCR, sequencing, and Western blotting.

**Transient transfections and drug treatments.** HEK293FT cells were transiently transfected using calcium phosphate method. U2OS cells were transiently transfected using Effectene Transfection Reagent (Qiagen). After 24 h of transfection, cells were treated with biotin (50 μM; Sigma-Aldrich) for the indicated exposure times. For stably transduced TRIPZ cell lines, induction with doxycycline (1 μg/ml; 24 h; Sigma-Aldrich) was performed prior to biotin treatment. ATO (1 μM; 2 h; Sigma-Aldrich) treatments were performed (with or without biotin, depending on experiment) prior to cell lysis or immunostaining.

**Western blot analysis.** Cells were washed 2× with PBS to remove excess biotin and lysed in highly stringent washing buffer 5 (WB5; 8 M urea, 1% SDS in 1X PBS) supplemented with 1× protease inhibitor cocktail (Roche) and 50 μM NEM.

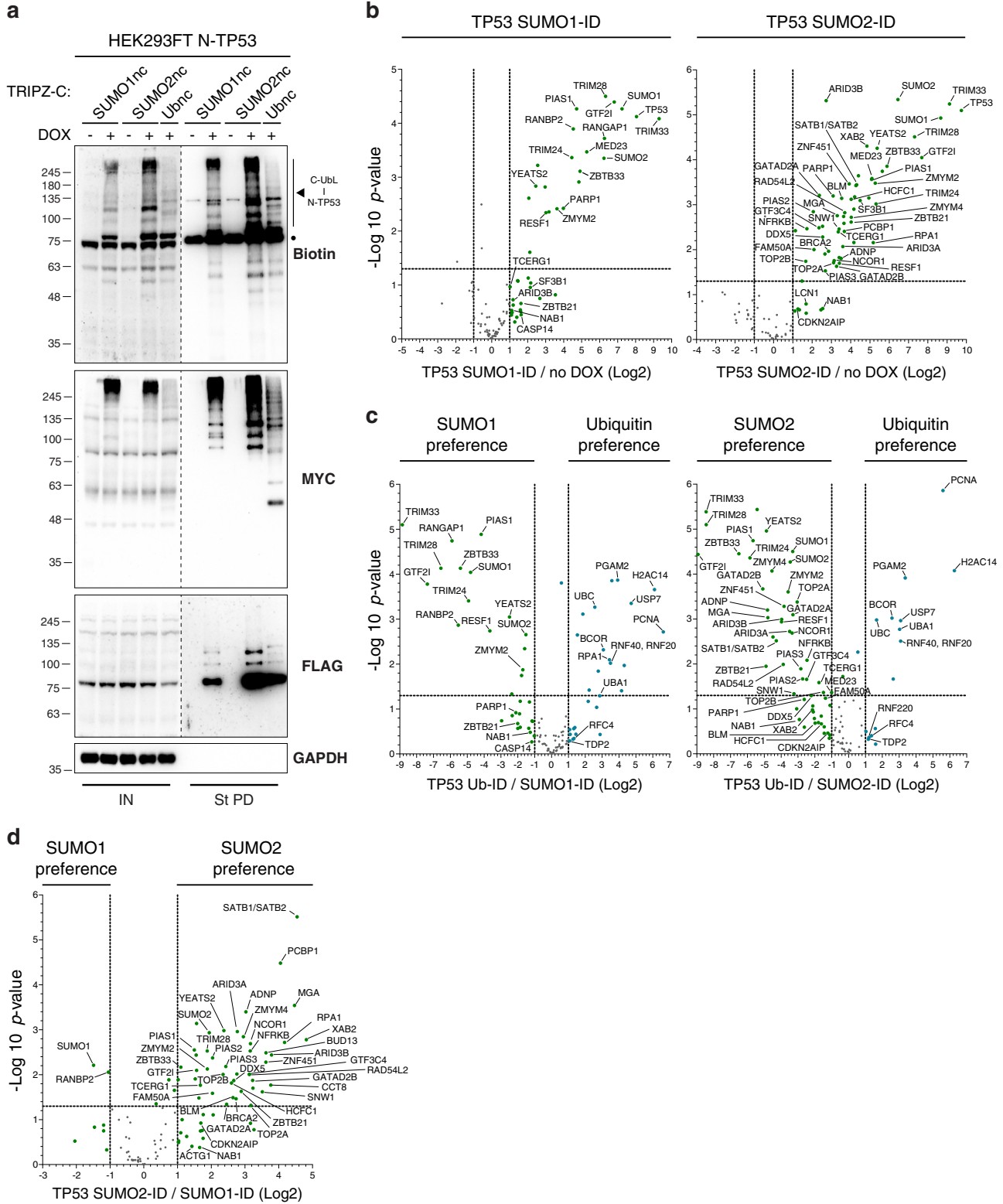

**Fig. 9 SUMO-ID identifies UbL and SUMO-paralogue preferential interactors of TP53. a** Western blot of HEK293FT double stable cell lines for FLAG-N-TP53 together with doxycycline-inducible TRIPZ-MYC-C-SUMO1nc, -SUMO2nc or -Ubnc. Doxycycline was added or not at 1 µg/mL for 24 h. 50 µM of biotin was added for 2 h. TP53 SUMO1-ID, SUMO2-ID and Ub-ID showed specific biotinylation patterns corresponding to each modification (black arrowhead). Dots indicate endogenous carboxylases that are biotinylated constitutively by the cell. Dotted lines indicate different exposures of the same blot. Results are representative of three independent pull-down experiments. Molecular weight markers are shown to the left of the blots in kDa. IN: input; St PD: streptavidin pull-down. Source data are provided in the Source Data file. Volcano plots of LC-MS analysis of (**b**) TP53 SUMO1-ID and SUMO2-ID, (**c**) SUMO1-ID or SUMO2-ID vs Ub-ID and (**d**) SUMO1-ID vs SUMO2-ID, from samples in **a**. SUMO-ID and Ub-ID of TP53 identified preferential interactors of each type of modification. Statistical analyses were performed by two-sided Student's *t* test. Data are provided as Supplementary Data 7.

Samples were then sonicated and cleared by centrifugation ($25,000 \times g$, 30 min, RT). 10–20 µg of protein was loaded for SDS-PAGE and transferred to nitrocellulose membranes. Blocking was performed in 5% milk in PBT (1× PBS, 0.1% Tween-20). Casein-based blocking solution was used for anti-biotin blots (Sigma). Primary antibodies were incubated over-night at 4 °C and secondary antibodies 1 h at room temperature (RT). Antibodies used: anti-biotin-HRP (1/1000; Cat#7075S), anti-Myc (1/1000; Cat#2276S), anti-alpha-Actinin (1/5000; Cat#6487S) (Cell Signaling Technology); anti-Flag (1/1000; Cat#F1804), anti-GTF2I (1/1000; Cat#HPA026638) (Sigma-Aldrich); anti-BirA (1/1000; Cat#11582-T16; SinoBiological); Proteintech antibodies: anti-IRF2BP2 (1/1000; Cat#18847-1-AP), anti-UBC9 (1/1000; Cat#14837-1-AP), anti-TRIM24 (1/1000; Cat#14208-1-AP), anti-TRIM33 (1/1000; Cat#55374-1-AP), anti-PIAS2 (1/1000; Cat#16074-1-AP), anti-PIAS4 (1/1000; Cat#14242-1-AP), anti-GATAD2B (1/1000; Cat#25679-1-AP), anti-MTA2 (1/1000; Cat#17554-1-AP), anti-RBBP4 (1/1000; Cat#20364-1-AP), anti-PML (1/1000; Cat#21041-1-AP), anti-GAPDH (1/5000; Cat#60004-1-Ig), anti-beta-Actin (1/5000; Cat#66009-1-Ig), anti-alpha-Tubulin (1/5000; Cat#66031-1-Ig); anti-PML (1/1000; Cat#A301-167A) (Bethyl); anti-GFP (1/1000; Cat#sc-8334) (SantaCruz); anti-Mouse-HRP (1/5000; Cat#115-035-062), anti-Rabbit-HRP (1/5000; Cat#111-035-045) (Jackson ImmunoResearch). Proteins were detected using Clarity ECL (BioRad) or Super Signal West Femto (ThermoFisher). Quantification of bands was performed using ImageJ (v2.0.0-rc-69/1.52n) software and normalized against loading/expression controls. 2 way ANOVA with Tukey correction for multiple comparisons was performed and the data is available in the Source Data file. All uncropped blots are provided within the Source Data file.

**Immunostaining and confocal microscopy**. U2OS and HEK293FT cells were seeded on 11 mm coverslips (25,000 cells per well; 24 well plate). HEK293FT coverslips were pre-treated with poly-L-lysine. After washing 3 times with 1× PBS, cells were fixed with 4% PFA supplemented with 0.1% Triton X-100 in PBS for 15 min at RT. Then, coverslips were washed 3 times with 1× PBS. Blocking was performed for 30 min at RT in blocking buffer (2% fetal calf serum, 1% BSA in 1× PBS). Primary antibodies were incubated for 2 h at 37 °C and cells were washed with 1× PBS 3 times. Primary antibodies used: anti-BirA (1/500; Cat#11582-T16; SinoBiological); anti-Myc (1/200; Cat#2276S; Cell Signaling Technology); anti-GTF2I (1/100; Cat#HPA026638; Sigma-Aldrich); Proteintech antibodies: anti-IRF2BP2 (1/100; Cat#18847-1-AP), anti-UBC9 (1/100; Cat#14837-1-AP), anti-TRIM24 (1/100; Cat#14208-1-AP), anti-TRIM33 (1/100; Cat#55374-1-AP), anti-PIAS2 (1/100; Cat#16074-1-AP), anti-PIAS4 (1/100; Cat#14242-1-AP), anti-CBX4 (1/100; Cat#18544-1-AP); anti-B23 (1/100; Cat#sc-271737; Santa Cruz); anti-SC35 (1/100; Cat#66000-1-Ig), anti-SUMO2 (1/100; Cat#SUMO-2 8A2; DSHB). Then secondary antibodies (together with fluorescent streptavidin) were incubated for 1 h at 37 °C, followed by nuclear staining with DAPI (10 min, 300 ng/ml in PBS; Sigma Aldrich). Secondary antibodies (ThermoFisher) were all used at 1/200: anti-Rabbit Alexa Fluor 488 (Cat#A-11034), anti-Mouse Alexa Fluor 568 (Cat#A-11031), anti-Rabbit Alexa Fluor 568 (Cat#A-11036). Streptavidin Alexa Fluor 594 (1/200; Cat#016-290-084; Jackson ImmunoResearch) was used. Fluorescence imaging was performed using confocal microscopy (Leica SP8 Lightning) with 63x Plan ApoChromat NA1.4. To obtain the signal histograms for co-localization studies in Fig. 5, we used the plot profile tool in ImageJ (v2.0.0-rc-69/1.52n). Colocalization was further confirmed using the Colocalization_Colormap and JACoP (Just Another Colocalization Plugin) plugins in ImageJ and the Coloc 2 tool in FIJI. Values were calculated using autothreshold or the Costes' automatic threshold options[72], point spread function = 3 and number of interactions = 20.

**PML NBs comparison**. The mean area and the number of PML NBs per cell of 40 U2OS cells stably expressing EFS–FLAG-NTurboID[194]–PMLIVa$^{WT}$ or -PMLIVa$^{3MAS}$ were analyzed in FIJI. After removing outliers (ROUT method, Q = 1%), two-sided unpaired $t$-test with Welch's correction was applied. Data are available in the Source Data file.

**Pulldown of biotinylated proteins**. Cleared lysates from WB5 lysis buffer were adjusted to the same protein concentration before incubating them with 1/50 ($v_{beads}/v_{lysate}$) equilibrated NeutrAvidin-agarose beads (ThermoFisher) over-night at RT. Due to the high-affinity interaction between biotin and streptavidin, beads were subjected to stringent series of washes, using the following WBs ($v_{WB}/2v_{lysate}$), all made in 1× PBS: 2× WB1 (8 M urea, 0.25% SDS); 3× WB2 (6 M Guanidine-HCl); 1× WB3 (6.4 M urea, 1 M NaCl, 0.2% SDS); 3× WB4 (4 M urea, 1 M NaCl, 10% isopropanol, 10% ethanol and 0.2% SDS); 1× WB1; 1× WB5; and 3× WB6 (2% SDS). Biotinylated proteins were eluted in 80 µl of Elution Buffer (4× Laemmli buffer, 100 mM DTT) through heating at 99 °C for 5 min and subsequent vortexing. Beads were separated by centrifugation ($25,000 \times g$, 2 min).

**Liquid chromatography-mass spectrometry (LC-MS/MS)**. A stable TRIPZ–MYC-CTurboID[195]–SUMO2nc U2OS cell line was transduced with either EFS–FLAG-NTurboID[194]–PMLIVa$^{WT}$ or EFS–FLAG-NTurboID[194]–PMLIVa$^{3MAS}$ for PML SUMO-ID experiments. Selection in blasticidin (5 µg/ml) and puromycin (1 µg/ml) was performed. Expression was validated by Western blot and immunostaining prior to scale-up for mass spectrometry. The TurboID-PML experiments used U2OS stable cell lines expressing low and equivalent levels of

PMLIVa$^{WT}$–TurboID, PMLIVa$^{3MAS}$–TurboID and TurboID alone, selected by blasticidin (5 µg/ml), and treated or not with ATO for 2 h. For SALL1 SUMO-ID, a HEK293FT stable cell line expressing low levels of TRIPZ–MYC-CTurboID[195]–SUMO2nc (selected with puromycin, 1 µg/ml) was transiently transfected with EFS-FLAG-NTurboID[194]–SALL1$^{WT}$ or EFS-FLAG-NTurboID[194]–SALL1$^{\Delta SUMO}$ (a mutant carrying Lys>Arg mutations at K571, K592, K982, K1086). For TP53 SUMO1-ID, SUMO2-ID and Ub-ID experiments, stable TRIPZ–MYC-CTurboID[195]–SUMO1nc, -SUMO2nc or -Ubnc HEK293FT cell lines were transduced with EFS–FLAG-NTurboID[194]–TP53. Selection in blasticidin (5 µg/ml) and puromycin (1 µg/ml) was performed. Expression was validated by Western blot prior to scale-up for mass spectrometry. Three independent pulldown experiments ($8 \times 10^7$ cells per replicate, 8 ml of lysis) were analyzed by LC-MS/MS.

Samples eluted from the NeutrAvidin beads were separated in SDS-PAGE (50% loaded) and stained with Sypro Ruby (Invitrogen) according to manufacturer's instructions. Gel lanes were sliced into 3 pieces as accurately as possible to guarantee reproducibility. The slices were subsequently washed in milli-Q water. Reduction and alkylation were performed using dithiothreitol (10 mM DTT in 50 mM ammonium bicarbonate) at 56 °C for 20 min, followed by iodoacetamide (50 mM chloroacetamide in 50 mM ammonium bicarbonate) for another 20 min in the dark. Gel pieces were dried and incubated with trypsin (12.5 µg/ml in 50 mM ammonium bicarbonate) for 20 min on ice. After rehydration, the trypsin supernatant was discarded. Gel pieces were hydrated with 50 mM ammonium bicarbonate, and incubated overnight at 37 °C. After digestion, acidic peptides were cleaned with TFA 0.1% and dried out in a RVC2 25 speedvac concentrator (Christ). Peptides were resuspended in 10 µl 0.1% formic acid (FA) and sonicated for 5 min prior to analysis.

PML and TP53 samples were analyzed in a hybrid trapped ion mobility spectrometry – quadrupole time of flight mass spectrometer (timsTOF Pro with PASEF, Bruker Daltonics) coupled online to a nanoElute liquid chromatograph (Bruker). This mass spectrometer takes advantage of a scan mode termed parallel accumulation – serial fragmentation (PASEF). Sample (200 ng) was directly loaded in a 15 cm Bruker nanoelute FIFTEEN C18 analytical column (Bruker) and resolved at 400 nl/min with a 100 min gradient. Column was heated to 50 °C using an oven.

For the analysis of SALL1 samples peptides were eluted from stage-tips in a solvent composed of deionized water/acetonitrile/FA at a ratio of 50/50/0.1 v/v/v. Peptides were lyophilized and dissolved in solvent A composed of deionized water/FA at a ratio of 100/0.1 v/v and subsequently analyzed by on-line C18 nano-HPLC MS/MS with a system consisting of an Ultimate 3000 nano gradient HPLC system (ThermoFisher), and an Exploris 480 mass spectrometer (ThermoFisher). Fractions were loaded onto a cartridge precolumn (5 mm × ID 300 µm, C18 PepMap 100 A, 5 µm particles (ThermoFisher)), using solvent A at a flow of 10 µl/min for 3 min and eluted via a homemade analytical nano-HPLC column (50 cm × ID 75 µm; Reprosil-Pur C18-AQ 1.9 µm, 120 A; Dr. Maisch GmbH). The gradient was run from 2% to 40% solvent B (water/acetonitrile/FA at a ratio of 20/80/0.1 v/v/v) in 40 min. The nano-HPLC column was drawn to a tip of ∼10 µm and acted as the electrospray needle of the MS source. The temperature of the nano-HPLC column was set to 50 °C (Sonation GmbH). The mass spectrometer was operated in data-dependent MS/MS mode for a cycle time of 3 s, with a HCD collision energy at 28 V and recording of the MS2 spectrum in the orbitrap, with a quadrupole isolation width of 1.6 Da. In the master scan (MS1) the resolution was set to 60,000, and the scan range was set to 300–1600, at an Automatic Gain Control (AGC) target of $3 \times 10^6$ with automated fill time. A lock mass correction on the background ion m/z = 445.12 was used. Precursors were dynamically excluded after $n = 1$ with an exclusion duration of 30 s, and with a precursor range of 10 ppm. Charge states 2–6 were included. For MS2 the first mass was set to 120 Da, and the MS2 scan resolution was 30,000 at an AGC target of 75,000 with automated fill time.

**Mass spectrometry data analysis**. Raw MS files were analyzed using MaxQuant (version 1.6.17)[73] matching to a human proteome (Uniprot filtered reviewed *H. sapiens* proteome, UP000005640 [https://www.uniprot.org/uniprot/?query=proteome:UP000005640%20reviewed:yes]) with a maximum of 4 missed cleavages and with precursor and fragment tolerances of 20 ppm and 0.05 Da. Label-Free Quantification was enabled with default values except for a ratio count set to 1. Slices corresponding to same lanes were considered as fractions. Biotinylation on lysine and on protein N-term was included as a variable modification for SALL1 SUMO-ID data, and biotinylated peptides were set to be included for quantification. Matching between runs and matching unidentified features were enabled. Only proteins identified with at least one peptide at FDR < 1% were considered for further analysis. Data were loaded onto the Perseus platform (version 1.6.14)[74] and further processed (Log2 transformation, imputation, median normalization when needed). A two-sided Student's $t$ test was applied in order to determine the statistical significance of the differences detected. Data were loaded into GraphPad Prism 8 version 8.4.3 to build the corresponding volcano-plots. Proteins detected with at least 2 peptides (except when otherwise specified) and in at least 2 of the 3 replicates were included. High confidence hits are considered when differences in Log2 of LFQ intensities are higher than 1 or statistically significant ($p < 0.05$).

Network analysis was performed using the STRING app version 1.4.2[75] in Cytoscape version 3.7.2[76], with a high-confidence interaction score (0.7). Transparency and width of the edges were continuously mapped to the String score (text mining, databases, coexpression, experiments, fusion, neighborhood and cooccurrence). The Molecular COmplex DEtection (MCODE) plug-in version 1.5.1[77] was used to identify highly connected subclusters of proteins (degree cutoff of 2; Cluster finding: Haircut; Node score cutoff of 0.2; K-Core of 2; Max. Depth of 100). Gene ontology analysis was performed using g:Profiler web server version e104_eg51_p15_3922dba[78].

**SIM enrichment analysis.** A thousand lists with the same number of proteins as PML SUMO-ID list (59) have been randomly generated from the human proteome (Uniprot filtered reviewed *H. sapiens* proteome, UP000005640). All those lists have been analyzed by adapting the script from[79] and running it on Python version 2.7.5, to obtain the content and number of SIM motifs per protein ($\psi$-$\psi$-X-$\psi$; $\psi$-X-$\psi$-$\psi$; $\psi$-$\psi$-$\psi$; where $\psi$ is either a L, I or V and X is any amino acid) and the number of SIMs per thousand of amino acids (STAA). After removing three outliers (lists 46, 782, 794; ROUT method, Q = 1%), STAA values from the random lists were normalized to Log2 and validated for Gaussian distribution (d'Agostino and Pearson normality test). Enrichments were computed using R software v3.6.0 and calculated as the ratio between PML SUMO-ID STAA value and the median of STAA values from the random lists. Empirical *p* values have been calculated by counting the number of random lists whose STAA values were as extreme as the PML SUMO-ID STAA value. The raw data from the SIM enrichment analysis and the script can be found in the Source Data file.

**Reporting summary.** Further information on research design is available in the Nature Research Reporting Summary linked to this article.

## Data availability

All data supporting the findings are provided within the paper, the Supplementary Data, Source Data file and the Supplementary Information. The crystal structure of *E. coli* BirA[80] is available under PDB entry number 1HXD. The fasta file of the human proteome (Uniprot filtered reviewed *H. sapiens* proteome, UP000005640) was downloaded from Uniprot. In addition, the mass spectrometry proteomics raw data have been deposited to the ProteomeXchange Consortium via the PRIDE partner repository[81] with the following dataset identifiers: PML SUMO-ID data, accession code PXD021770, PML-TurboID data, accession code PXD021809, SALL1 SUMO-ID data, accession code PXD021923 and TP53 SUMO-ID/Ub-ID data, accession code PXD027759. Processed LC-MS/MS data as well as their corresponding gene ontology source data are provided as Supplementary Data files. Source data are provided with this paper.

## Code availability

The script for the SIM enrichment analysis can be found in the Source Data file.

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

## Acknowledgements

We are thankful to Iraide Escobes for her work in the Proteomics Platform at CIC bioGUNE and Arnoud de Ru in the Center for Proteomics and Metabolomics at the LUMC. O.B.-G., F.T., R.B. and A.C.O.V. acknowledge funding by the grant 765445-EU (UbiCODE Program). R.B. acknowledges funding by grants BFU2017-84653-P and PID2020-114178GB-I00 (MINECO/FEDER, EU), SEV-2016-0644 (Severo Ochoa Excellence Program), SAF2017-90900-REDT (UBIRed Program) and IT1165-19 (Basque Country Government). Additional support was provided by the Department of Industry, Tourism, and Trade of the Basque Country Government (Elkartek Research Programs) and by the Innovation Technology Department of the Bizkaia County. VM acknowledges FPI grant PRE2018-086230 (MINECO/FEDER, EU). F.E. is at Proteomics Platform, member of ProteoRed-ISCIII (PT13/0001/0027). F.E. and A.M.A. acknowledge CIBER-ehd. A.C. acknowledges the Basque Department of education (IKERTALDE IT1106-16), the MCIU (PID2019-108787RB-I00 (FEDER/EU), Severo Ochoa Excellence Accreditation SEV-2016-0644, Excellence Networks RED2018-102769-T), the AECC (GCTRA18006CARR), La Caixa Foundation (ID 100010434), under the agreement LCF/PR/HR17, and the European Research Council (Starting Grant 336343, PoC 754627, Consolidator grant 819242). CIBERONC was co-funded with FEDER funds. U.M. acknowledges the Basque Government Department of Education (IT1165-19) and the Spanish MCIU (SAF2016-76898-P (FEDER/EU)).

## Author contributions

O.B.-G., J.D.S. and R.B designed experiments, analyzed data and wrote the manuscript. O.B.-G., F.T., V.M., I.C., L.M.-C., A.R.C., C.P., M.A., I.I. and J.D.S. developed experimental protocols and performed experiments. A.C., A.M.A., F.E., U.M. and A.C.O.V. provided scientific resources.

## Competing interests
The authors declare no competing interests.
