## [Peer Review File · Nature Communications]

REVIEWER COMMENTS

Reviewer #1 (Remarks to the Author):

This work develops a split enzyme for promiscuous biotinylation of nearby proteins, dependent upon the two halves being tethered in proximity and the addition of biotin. This is then applied to identify SUMO-dependent protein interactors. Proximity labeling is a fast moving area with broad relevance to the study of protein signaling and I think that the work here should have wide interest, as well as facilitating the analysis of many other kinds of transient protein contacts. There is thorough exploration of interactors related to PML and SALL1. I felt that the work was carefully performed and was well written.

Major points:

I found it hard work keeping track of all the constructs and what all the arrows/squares are supposed to mean, so direct and simpler labeling or more cartoons would help the reader.

It is important to deposit the sequence of key inserts in GenBank to facilitate the reproduction of this work.

Minor points:

“Briefly, the C-terminal di-glycine motif of mature SUMOs binds lysines in substrates through the sequential action of E1 SUMO-activating enzyme SAE1/SAE2, E2 conjugating enzyme UBC9 and SUMO E3 ligases 2 .”

“Binds” may suggest non-covalent.

Spell out SENP at first use.

“Dots indicate endogenous carboxylases.” Explain somewhere that these are biotinylated naturally by the cell.

Figure 5 seems a non-ideal way to analyze colocalization. There are various quantitative methods that can study the degree of colocalization through a field of view and would give a clearer picture of what is found.

Reviewer #2 (Remarks to the Author):

In this paper, Barroso-Gomila et al. describe SUMO-ID, a novel approach to identify SUMO-dependent interaction partners of proteins of interest. To do so, the N-terminal part of the promiscuous biotin ligase TurboID is fused to a protein of interest, while the C-terminal part is fused to SUMO. When SUMO interacts covalently or non-covalently with the substrate of interest, as a result proximity protein partners that depend on this interaction will be biotinylated for downstream purification and identification by mass spectrometry. To develop their approach, the authors designed a new split-TurboID version, which is a valuable tool on its own. The authors first validated SUMO-ID following self-biotinylation of mainly PML by western blot and microscopy and then improved the approach by using non-cleavable versions of SUMO and doxycycline-inducible expression. They then applied SUMO-ID to identify SUMO-dependent interaction partners of PML and report 59 high-confident partners of which they validated several hits, confirming their localization to PML nuclear bodies (NBs). Comparing these hits to the regular proteome of PML revealed that the SUMO-dependent proteome constitutes a subset of the total PML proteome. Finally, the authors applied SUMO-ID to the poorly characterized SUMO substrate SALL1 and validated several newly identified hits of the NuRD complex.

The manuscript is well written and properly constructed, and the experimental data is of high quality. While the results on PML and PML NBs are relevant, especially in light of arsenic trioxide (ATO) treatment of acute myelocytic leukemia, I do have some major concerns about the broader applicability and versatility of the method that should be addressed.

Major comments

What are the distance constraints between the SUMOylation/SIM sites of interest and the substrate N-terminus? Clearly, the approach will only work if the two halves of the split-TurboID can find each other within the distance constraints provided by the linker sequences between C-TurboID and SUMO on the one hand and between NTurboID and the substrate on the other hand. In Figure 1A these linkers are drawn as very flexible regions, but in panel B they are not included in the construct design. These limitations of the approach should be addressed experimentally in order to present TurboID as a broadly applicable method. To this end, the authors could use for instance their well-characterized NTbID-PML3MAS construct in which they could re-introduce the K65, K160 or K490 sites and/or the SIM domain separately to look at site-specific differences.

Related to this, if one wants to use SUMO-ID to find binding partners of an uncharacterized SUMO substrate, what is the negative control that the authors advice to use? Should this always be a “SUMO-death” mutant in which all known/predicted sites are mutated like the authors did for PML and SALL1? At least the authors should guide the readers in the design of SUMO-ID experiments and the selection of appropriate control conditions as this might not be so trivial.

Finally, the authors do not present any MS data with SUMO1 and often SUMO1 seems to perform less well compared to SUMO2 on western blot. Can the authors comment on these differences? To support broad applicability of SUMO-ID, can the authors characterize a third SUMO substrate in combination with SUMO1? A comparative analysis with SUMO2 would also be most interesting.

Other comments:

1. Figure 2D: Flag and Myc control blots are missing and repeat experiments are required for proper quantification and statistics. Please directly compare the new T194/G195 split-TurboID with the previous E256/G257 split-TurboID in this experiment. Also, I suggest to discuss (and move up) these data before the SUMO-ID data in panel B and C.

2. Figure 3A: Flag control blot is missing and can the authors comment on the MW difference observed for SUMO1/2 and SUMO1/2 Δ G? One would expect that the Δ G variant runs a bit lower on the gel.

3. Figure 3D: the difference between true NB and NB-like bodies are clear, but differences in size or quantity of these bodies (e.g. after ATO treatment) should be quantified to support solid conclusions.

4. Figure 8C: there seems to be one hit specific for SALL1 Δ SUMO. Can the authors comment on this observation? Is this a SUMO-independent interaction?

5. For clarity, please be consistent in the annotation of the constructs used throughout the manuscript, figures and legends. For instance, avoid the mixed use of CTurboID-SUMO, CTbID-SUMO and C-SUMO.

Reviewer #3 (Remarks to the Author):

The authors describe a new mass spectrometry-based approach to identify protein-protein interactions that are mediated by sumoylation. The approach uses split-TurboID fusion proteins in which the N-terminus of the TurboID protein is fused to protein substrates of interest and the C-terminus of the TurboID protein is fused to SUMO. When expressed in cells in the presence of biotin, proteins that come into close proximity to the SUMO modified substrate are biotinylated and can be purified and identified by mass spectrometry. The authors validated the utility of the approach by identifying proteins that associate with sumoylated PML as well as the transcriptional repressor, SALL1.

It is generally believed that a major mechanism of action of sumoylation involves mediating associations between modified substrates and other proteins. Identifying the proteins that associate specifically with SUMO modified substrates can therefore provide valuable insights into function. This, however, is often challenging given the transient nature of the modification. The approach described in the manuscript is therefore valuable and it can be anticipated that others in the field will adopt the approach to explore the specific effects of SUMO on other proteins.

Overall, the manuscript is well written, the data are of high quality and experiments largely contain proper controls. A number of comments and suggestions for improving the manuscript include:

- 1) It is presumed that nuclear foci seen in Figures 2 and 3 are PML nuclear bodies, but it would be helpful to verify this by co-staining with PML or other marker protein.
- 2) With regard to foci in 2C, it is interesting that all of the C-SUMO1 and C-CUMO2 are recruited to the foci when co-expressed with N-PML (in comparison to expression without PML where they are throughout the nucleoplasm). Is this just an artefact of PML overexpression, or is it also possible that the TurboID tag interactions stabilize association with PML (and thereby SUMO modification)? In the

blot in 3B it appears that PML is more efficiently modified with C-SUMO1 compared to N-SUMO1. Does N-SUMO1 also concentrate in the foci to the same level as C-SUMO1? If the tag has an effect on modification efficiency, this may have implications on function.

3) Sumoylation is generally dynamic and it can be expected that this dynamic nature influences the assembly of factors and ultimate function. Thus, it is surprising that the MS analysis was done with the non-cleavable form of SUMO2 when the native form also appeared to be working. It would be helpful for the authors to rationalize this choice in more detail. Where efforts to identify interacting proteins using native forms of SUMO1 or SUMO2 attempted and not successful due to turnover, longer labeling times etc.?

4) It is also noted that MS analysis was only done using SUMO2. A comparison with SUMO1 could reveal interesting paralog-specific differences.

5) It is encouraging that tagged-SALL1 identifies a number of known interactors. However, functional transcription activity assays of tagged and untagged-SALL1 are really needed to know if there are possible effects of the tag that could impact the findings.

Reviewer #1 (Remarks to the Author):

This work develops a split enzyme for promiscuous biotinylation of nearby proteins, dependent upon the two halves being tethered in proximity and the addition of biotin. This is then applied to identify SUMO-dependent protein interactors. Proximity labeling is a fast moving area with broad relevance to the study of protein signaling and I think that the work here should have wide interest, as well as facilitating the analysis of many other kinds of transient protein contacts. There is thorough exploration of interactors related to PML and SALL1. I felt that the work was carefully performed and was well written.

We thank the reviewer for the positive comments.

Major points:

- I found it hard work keeping track of all the constructs and what all the arrows/squares are supposed to mean, so direct and simpler labeling or more cartoons would help the reader.

We added direct labelling on the blots, making explicit the bands corresponding to each N-substrates and their C-SUMO modifications. In the text, we changed the nomenclature to the more direct C-SUMO and N-substrate (N-PML, N-SALL1, etc). We also added a cartoon explaining the FRB/FKBP dimerization system.

- It is important to deposit the sequence of key inserts in GenBank to facilitate the reproduction of this work.

A full list of the constructs used in this work is available as Supplementary Table 1, and reference to those is highlighted in the text. Those constructs will be available in Addgene. Furthermore, we added the sequences of the 15 most representative constructs in the Source data section. Additional details about other constructs will be available upon request.

Minor points

- “Briefly, the C-terminal di-glycine motif of mature SUMOs binds lysines in substrates through the sequential action of E1 SUMO-activating enzyme SAE1/SAE2, E2 conjugating enzyme UBC9 and SUMO E3 ligases 2 .”
“Binds” may suggest non-covalent.

We totally agree that the use of “Binds” might lead to confusion. We thus changed the sentence to: “Briefly, the C-terminal di-glycine motif of mature SUMOs mediates modification of target lysines in substrates through the sequential action of E1 SUMO-activating enzyme SAE1/SAE2, E2 conjugating enzyme UBC9 and SUMO E3 ligases.”

- Spell out SENP at first use.

We spelled out SENP at first use: Sentrin-specific proteases.

- “Dots indicate endogenous carboxylases.” Explain somewhere that these are biotinylated naturally by the cell.

We added a sentence in the figure legends when referring to the carboxylases, explaining that endogenous carboxylases are biotinylated constitutively by the cell.

- Figure 5 seems a non-ideal way to analyze colocalization. There are various quantitative methods that can study the degree of colocalization through a field of view and would give a clearer picture of what is found.

Upon Reviewer's request, we performed the colocalization analysis using the Colocalization_Colormap and JACoP (Just Another Colocalization Plugin) plugins in ImageJ and the Coloc 2 tool in FIJI. Details on the method are included in the "Methods" section and the new information has been added as a Supplementary Fig. 8.

Reviewer #2 (Remarks to the Author):

In this paper, Barroso-Gomila et al. describe SUMO-ID, a novel approach to identify SUMO-dependent interaction partners of proteins of interest. To do so, the N-terminal part of the promiscuous biotin ligase TurboID is fused to a protein of interest, while the C-terminal part is fused to SUMO. When SUMO interacts covalently or non-covalently with the substrate of interest, as a result proximity protein partners that depend on this interaction will be biotinylated for downstream purification and identification by mass spectrometry. To develop their approach, the authors designed a new split-TurboID version, which is a valuable tool on its own. The authors first validated SUMO-ID following self-biotinylation of mainly PML by western blot and microscopy and then improved the approach by using non-cleavable versions of SUMO and doxycycline-inducible expression. They then applied SUMO-ID to identify SUMO-dependent interaction partners of PML and report 59 high-confident partners of which they validated several hits, confirming their localization to PML nuclear bodies (NBs). Comparing these hits to the regular proteome of PML revealed that the SUMO-dependent proteome constitutes a subset of the total PML proteome. Finally, the authors applied SUMO-ID to the poorly characterized SUMO substrate SALL1 and validated several newly identified hits of the NuRD complex.

The manuscript is well written and properly constructed, and the experimental data is of high quality. While the results on PML and PML NBs are relevant, especially in light of arsenic trioxide (ATO) treatment of acute myelocytic leukemia, I do have some major concerns about the broader applicability and versatility of the method that should be addressed.

We thank the reviewer for the positive comments.

Major comments

- What are the distance constraints between the SUMOylation/SIM sites of interest and the substrate N-terminus? Clearly, the approach will only work if the two halves of the split-TurboID can find each other within the distance constraints provided by the linker sequences between C-TurboID and SUMO on the one hand and between NTurboID and the substrate on the other hand. In Figure 1A these linkers are drawn as very flexible regions, but in panel B they are not included in the construct design. These limitations of the approach should be addressed experimentally in order to present TurboID as a broadly applicable method. To this end, the authors could use for instance their well-characterized

NTbID-PML3MAS construct in which they could re-introduce the K65, K160 or K490 sites and/or the SIM domain separately to look at site-specific differences.

We agree with the Reviewer that there might be some distance constraints that may impact on the efficiency of SUMO-ID. The GSQ flexible linker located between the C-terminal Turbo ID (C) and SUMOs and the N-terminal TurboID (N) and the substrate might overcome this issue to a certain extent. We included the GSQ linker in the construct schematic design in Fig. 1b.

Following the Reviewers' suggestion, we re-introduced the K65 or the K490 residues into the N-PML^{3MAS} construct. We performed SUMO-ID by transiently transfecting HEK293FT-TRIPZ-C-SUMO2nc cell line with N-PML^{WT}, N-PML^{2MAS} (K65), N-PML^{2MAS} (K490) or N-PML^{3MAS} (see Figure 1 of this rebuttal letter below). As in previous experiments, we observed a strong SUMO-ID activity when using PML^{WT}. Long exposure times showed some low intensity specific biotinylation pattern when using N-PML^{2MAS} (K65) or N-PML^{2MAS} (K490) compared to N-PML^{3MAS} (arrowhead), suggesting that SUMOylation and reconstitution of TurboID is happening at those specific lysines. However, the SUMO-ID activity in the K65 or K490 re-introduced constructs was not much higher compared to the background obtained with the N-PML^{3MAS}. This might be due to the conformational consequences that each of the SUMO sites might have in the 3D structure of PML, and it is unclear whether each SUMO site can contribute to the SUMOylation of other sites in PML. In addition, whether those PML^{2MAS} form true nuclear bodies is not known, which might also affect to PML SUMOylation and thus SUMO-ID efficiency. Furthermore, we observed that the stability of the N-PML^{2MAS} (K490) version is lower than the N-PML^{2MAS} (K65) or N-PML^{3MAS} counterparts. This might be due to the fact that K490 is also a ubiquitylation site (Akimov et al., 2018; DOI: 10.1038/s41594-018-0084-y) and may regulate PML stability.

Figure 1: Re-introduction of K65 and K490 into PML^{3MAS} shows specific but low intensity SUMO-ID.

Although specific SUMO-ID was detected at K65 and K490, the issue of the distance between the SUMOylation site and the N-terminus of the specific substrate, as well as the contribution of the GSQ linker, still remains unclear.

To address this issue in another way, we generated versions of TRIPZ-C-SUMO2nc and N-PML^{WT} that lack the GSQ linker (direct fusions; Figure 2 of this rebuttal letter). We performed transient SUMO-ID experiments on HEK293FT cells, combining each of the 4 constructs. Surprisingly, no clear differences of the SUMO-ID efficiency were observed among the different combinations. This suggests that the SUMO sites in PML are close enough to the N-terminus for the TurboID enzyme to refold, even without the GSQ linker. In conclusion, these experiments did not allow us to conclude the relevance of the linker in PML, probably due to the specific nature of PML. In fact, PML may not be the best substrate to ask these questions, since it forms dimers and higher-order parallel and anti-parallel assemblies in forming nuclear bodies, so cis- vs. trans-events might be mixed. A comment on the suitability to check different linker lengths was included in the “Discussion” section.

Figure 2: Removal of GSQ linker does not affect PML SUMO-ID activity.

- Related to this, if one wants to use SUMO-ID to find binding partners of an uncharacterized SUMO substrate, what is the negative control that the authors advice to use? Should this always be a “SUMO-death” mutant in which all known/predicted sites are mutated like the authors did for PML and SALL1? At least the authors should guide the readers in the design of SUMO-ID experiments and the selection of appropriate control conditions as this might not be so trivial.

We believe that the optimal negative control of a SUMO-ID experiment would be the “SUMO-dead” mutant, as shown for PML and SALL1. High-throughput SUMO site mapping through proteomics by multiple research groups has yielded candidate sites for a majority of targets, and K>R substitutions of the major site(s) will yield a useful experimental control. Otherwise, ΔGG versions of C-SUMOs

would also represent a good negative control, although they could interact through SIMs present in the substrate. Another approach could be to use a non-induced, DOX negative control when the expression of C-SUMOs is in an inducible vector, as shown for the new TP53 SUMO-ID experiments in the new Fig. 9 of the manuscript. In addition, when different UbL-IDs are performed, one could represent the control of the other (e.g. SUMO2 VS Ubiquitin), and this is particularly interesting to see UbL specificity, as shown for TP53 in Fig. 9. Comments on the proper controls during the design of SUMO-ID experiments have been added in the “Discussion” section.

- Finally, the authors do not present any MS data with SUMO1 and often SUMO1 seems to perform less well compared to SUMO2 on western blot. Can the authors comment on these differences? To support broad applicability of SUMO-ID, can the authors characterize a third SUMO substrate in combination with SUMO1? A comparative analysis with SUMO2 would also be most interesting.

As the Reviewer noticed, SUMO2 seems to perform more efficiently by western blot compared to SUMO1. This is probably due to the fact that SUMO2 isoform is well characterized as a “chain builder” isoform, while SUMO1 is considered a “chain terminator”. In the case of PML, we would thus expect that more molecules of C-SUMO2 than of C-SUMO1 would modify the same N-PML, increasing thus the probabilities of an efficient SUMO-ID biotinylating activity compared to SUMO1.

As pointed out, no SUMO1-ID data were presented in the manuscript. Following the Reviewer’s suggestion, we performed SUMO1-ID and SUMO2-ID on a new substrate. We also added a Ub-ID condition to see the broad applicability of the system. We chose TP53 as a new substrate, as it is well-studied with characterized interactors, and known to be modified by SUMO1, SUMO2 and Ubiquitin. Comparisons between TP53 SUMO1-ID, SUMO2-ID and Ub-ID identified preferential interactors for each type of modification, confirming the high specificity of the technique. These results are in the new Fig. 9 in the manuscript.

Other comments

- Figure 2D: Flag and Myc control blots are missing and repeat experiments are required for proper quantification and statistics. Please directly compare the new T194/G195 split-TurboID with the previous E256/G257 split-TurboID in this experiment. Also, I suggest to discuss (and move up) these data before the SUMO-ID data in panel B and C.

Following the Reviewer’s suggestion, quantification and statistics were performed over repetitions of the experiment shown in Fig. 2D (see new Fig. 2d). Raw data of the quantification and the statistical tests performed were added as a Source Data file. We also added the MYC and the FLAG control blots, that fit nicely with the BirA blot.

While we appreciate the suggestion, we believe that direct comparison of the T194/G195 with the previous E256/G257 split-point using the FRB/FKBP dimerization system would not provide additional information. When we realized that the NTurboID²⁵⁶ is active by itself and biotinylates without the need of CTurboID²⁵⁷ (Supplementary Fig. 1), we stopped using this split-point as it not suitable for reconstitution-dependent experiments. This was also observed by Cho and co-workers in their recent report on Split-TurboID (Cho et al., 2020; DOI: 10.1073/pnas.1919528117). They use the FRB/FKBP system to show that E256/G257 split-TurboID biotinylating activity is not fully dependent on reconstitution of TurboID, and that the biotinylation activity of the fully folded enzyme is not very efficient. We prefer to maintain the established order of data presentation to emphasize more in the applicability of the SUMO-ID strategy.

- Figure 3A: Flag control blot is missing and can the authors comment on the MW difference observed for SUMO1/2 and SUMO1/2 Δ GG? One would expect that the Δ GG variant runs a bit lower on the gel.

FLAG control blot was added in Fig. 3a.

There are in fact differences on size of SUMO1/2 and their respective Δ GG variants: Δ GG modification of SUMO1/2 makes them not processable by SENPs, and they thus maintain their native state. This means that they will maintain the HSTV "SUMO tail" that is not removed by SENPs. They are thus higher in MW compared to their WT counterparts, and they migrate more slowly in a gel. Explanation of this has been added in the main text.

- Figure 3D: the difference between true NB and NB-like bodies are clear, but differences in size or quantity of these bodies (e.g. after ATO treatment) should be quantified to support solid conclusions.

We observed that true PML NBs were more abundant and much smaller in area than the NB-like bodies formed by PML^{3MAS}. As suggested by the Reviewer, we performed the quantification and statistical analysis of size and number of the NBs to support our conclusions. The new results are shown in Supplementary Fig. 7b, methods have been explained in the "Methods" section, and the raw data were added as a Source Data file.

- Figure 8C: there seems to be one hit specific for SALL1 Δ SUMO. Can the authors comment on this observation? Is this a SUMO-independent interaction?

As the Reviewer observed, there is a highly enriched hit in SALL1 Δ SUMO compared to the WT: BAZ1B. The hit has been now highlighted in the Fig. 8c and discussed in the main text. BAZ1B is a tyrosine-protein kinase that acts as a transcriptional regulator. Interestingly, it is highly SUMOylated with at least 5 putative SUMO sites identified (K826, K853, K1043, K1089 and K1107) (Hendriks et al., 2017; DOI: 10.1038/nsmb.3366). Our data suggest that SUMOylated BAZ1B might interact with SALL1 and that this interaction might be enhanced when SALL1 SUMOylation is inhibited. It would be of great interest to study whether SALL1 is phosphorylated by BAZ1B, and if that interaction is SUMO-SIM interaction dependent, which could be the subject of a new monographic study. The other hit that seems to be enriched for SALL1 Δ SUMO is CPS1. This protein is an enzyme involved in the urea cycle, by removing the excess of ammonia from the cells.

- For clarity, please be consistent in the annotation of the constructs used throughout the manuscript, figures and legends. For instance, avoid the mixed use of CTurboID-SUMO, CTbID-SUMO and C-SUMO.

As recommended, we used now C-SUMOs and N-substrate abbreviations all the way through the manuscript, making it clear that those nomenclatures refer to CTurboID-SUMO and NTurboID-substrate when first used.

Reviewer 3:

The authors describe a new mass spectrometry-based approach to identify protein-protein interactions that are mediated by sumoylation. The approach uses split-TurboID fusion proteins in which the N-terminus of the TurboID protein is fused to protein substrates of interest and the C-terminus of the TurboID protein is fused to SUMO. When expressed in cells in the presence of biotin, proteins that come into close proximity to the SUMO modified substrate are biotinylated and can be

purified and identified by mass spectrometry. The authors validated the utility of the approach by identifying proteins that associate with sumoylated PML as well as the transcriptional repressor, SALL1.

It is generally believed that a major mechanism of action of sumoylation involves mediating associations between modified substrates and other proteins. Identifying the proteins that associate specifically with SUMO modified substrates can therefore provide valuable insights into function. This, however, is often challenging given the transient nature of the modification. The approach described in the manuscript is therefore valuable and it can be anticipated that others in the field will adopt the approach to explore the specific effects of SUMO on other proteins.

Overall, the manuscript is well written, the data are of high quality and experiments largely contain proper controls.

We thank the reviewer for the positive comments.

A number of comments and suggestions for improving the manuscript include:

- 1) It is presumed that nuclear foci seen in Figures 2 and 3 are PML nuclear bodies, but it would be helpful to verify this by co-staining with PML or other marker protein.

Following the Reviewer's suggestion, we transfected the U2OS YFP-PML Knock-In cell line with N-PML and checked co-localization of BirA and YFP by confocal microscopy. This figure was added as Supplementary Fig. 7a. Indeed, we observed co-localization of N-PML and endogenous YFP-PML at nuclear bodies, showing that BirA-positive N-PML associates with true PML nuclear bodies.

- 2) With regard to foci in 2C, it is interesting that all of the C-SUMO1 and C-CUMO2 are recruited to the foci when co-expressed with N-PML (in comparison to expression without PML where they are throughout the nucleoplasm). Is this just an artefact of PML overexpression, or is it also possible that the TurboID tag interactions stabilize association with PML (and thereby SUMO modification)? In the blot in 3B it appears that PML is more efficiently modified with C-SUMO1 compared to N-SUMO1. Does N-SUMO1 also concentrate in the foci to the same level as C-SUMO1? If the tag has an effect on modification efficiency, this may have implications on function.

As the polyclonal BirA antibody recognizes both, the NTurboID¹⁹⁴ and the CTurboID¹⁹⁵, most of the immunofluorescence intensities observed in the green channel in Fig. 2c correspond to the prominent PML nuclear bodies when both C-SUMO and N-PML are expressed. However, increasing the intensities in that channel enables to see the nucleoplasmic staining of C-SUMOs (see Figure 3 of this rebuttal letter below).

As for the second part of this point, we understand that the Reviewer is referring to Fig. 2b. In Fig. 2b, FLAG panel, C-SUMO1 appears to modify more efficiently N-PML than N-SUMO1. This could be related to the size of the tag, as NTurboID¹⁹⁴ is bigger than CTurboID¹⁹⁵ and might influence the efficiency of substrate modification. This is the main reason why we choose to tag SUMO with CTurboID¹⁹⁵.

U2OS N – PML^{WT}

Figure 3: Increasing intensities of BirA enables to detect nucleoplasmic C-SUMO staining.

- 3) Sumoylation is generally dynamic and it can be expected that this dynamic nature influences the assembly of factors and ultimate function. Thus, it is surprising that the MS analysis was done with the non-cleavable form of SUMO2 when the native form also appeared to be working. It would be helpful for the authors to rationalize this choice in more detail. Where efforts to identify interacting proteins using native forms of SUMO1 or SUMO2 attempted and not successful due to turnover, longer labeling times etc.?

The native forms of SUMO1 and SUMO2 also showed efficient SUMO-ID activity. We choose to develop SUMO-ID with non-cleavable forms of SUMO1 and SUMO2 for 3 main reasons:

- 1- To avoid specificity issues derived from recycling of pre-labelled C-SUMOs: when using native SUMO1 and SUMO2 versions, we observed a non-negligible amount of free biotinylated C-SUMOs (Fig. 2b, empty squares); those can incorporate into other substrates and lead to non-specific identification. Furthermore, when we performed a first MS pilot experiment with N-PML together with the native form of C-SUMO1, the majority of peptides we got corresponded to RanGAP1, the most SUMO1 modified substrate. This suggested that more transient SUMOylation of other substrates was lost by the recycling.
- 2- To avoid the interaction of free C-SUMOs with SIMs present in the substrate: as evidenced in Fig. 3a, C-SUMOs can interact with SIMs present in the substrate when they are in their free unincorporated state (SUMO^{AGG}). Using non-cleavable versions of C-SUMOs showed that all of them are incorporated into the substrates, which enabled us to avoid that source of non-specificity.
- 3- To diminish biotin labelling times while having efficient SUMO-ID biotinylation activity: SUMOylation is a very transient modification, with high dynamics. Using non-cleavable versions of C-SUMOs enable us to gain in efficiency of SUMO-ID while decreasing biotin labelling times, as C-SUMOs will remain much longer into the substrate. Decreasing biotin labelling times is crucial to gain specificity, as longer incubation timings would likely enhance the labelling radius and contribute to non-specificity.

A detailed explanation of the choice has been now added in the main text.

- 4) It is also noted that MS analysis was only done using SUMO2. A comparison with SUMO1 could reveal interesting paralog-specific differences.

We also agree that comparing SUMO1-ID and SUMO2-ID would be of great interest to identify paralog-preferential interactors if they exist. We thus decided to perform SUMO1-ID and SUMO2-ID on a new substrate, as suggested by another Reviewer. We also added a Ub-ID condition to see the broader applicability of the system. We chose TP53 as a new substrate, as it is well-studied with characterized interactors, and known to be modified by SUMO1, SUMO2 and Ubiquitin. Comparisons between TP53 SUMO1-ID, SUMO2-ID and Ub-ID identified preferential interactors for each type of modification, and a new Fig. 9 was added in the manuscript.

- 5) It is encouraging that tagged-SALL1 identifies a number of known interactors. However, functional transcription activity assays of tagged and untagged-SALL1 are really needed to know if there are possible effects of the tag that could impact the findings.

SALL1 targets are poorly characterized in the literature. Chi et al found that SALL1 overexpression is associated with downregulation of C-MYC and upregulation of p21 in human glioblastoma cell lines (Chi et al., 2019; DOI: 10.12659/MSMBR.915067). We tried to validate such effect in HEK293FT cells by transiently transfecting untagged SALL1 and N-SALL1, and checking RNA levels of C-MYC and p21 by RT-qPCR (see Figure 4 of this rebuttal letter below). We could not confirm the downregulation of C-MYC by overexpression of SALL1 in HEK293FT cells. However, overexpression of both untagged SALL1 and N-SALL1 resulted in upregulation of p21. This result shows that the NTurboID¹⁹⁴ tag on SALL1 does not drastically affect its transcriptional regulation activity.

Figure 4: NTurboID¹⁹⁴ tag does not affect SALL1 transcriptional activity.

REVIEWERS' COMMENTS

Reviewer #2 (Remarks to the Author):

I appreciate the efforts of the authors to experimentally address my main comments and to clarify certain points in the main text. The manuscript has now substantially improved and can be accepted for publication.

Francis Impens

Reviewer #3 (Remarks to the Author):

The authors have addressed all of the major concerns with addition of new data or clarifications added to the text. Overall, the manuscript is significantly improved and conclusions are supported by the findings.